# Cerebrospinal fluid concentration of complement component 4A is increased in first episode schizophrenia

Jessica Gracias[1], Funda Orhan[1], Elin Hörbeck [2,3], Jessica Holmén-Larsson[2], Neda Khanlarkani[1], Susmita Malwade[1], Sravan K. Goparaju[1], Lilly Schwieler[1], İlknur Ş. Demirel[1], Ting Fu[4], Helena Fatourus-Bergman[5,6], Aurimantas Pelanis[7], Carleton P. Goold[8], Anneli Goulding[3,9], Kristina Annerbrink[3], Anniella Isgren[2,3], Timea Sparding[2], Martin Schalling [10], Viviana A. Carcamo Yañez[11], Jens C. Göpfert[11], Johanna Nilsson [2], Ann Brinkmalm [2], Kaj Blennow [2,12], Henrik Zetterberg [2,12,13,14,15], Göran Engberg [1], Fredrik Piehl [5], Steven D. Sheridan [4], Roy H. Perlis[4], Simon Cervenka [5,6,16], Sophie Erhardt [1], Mikael Landen [2,17] & Carl M. Sellgren [1,6] ✉

Postsynaptic density is reduced in schizophrenia, and risk variants increasing complement component 4A (*C4A*) gene expression are linked to excessive synapse elimination. In two independent cohorts, we show that cerebrospinal fluid (CSF) C4A concentration is elevated in patients with first-episode psychosis (FEP) who develop schizophrenia (FEP-SCZ: median 0.41 fmol/ul [CI = 0.34–0.45], FEP-non-SCZ: median 0.29 fmol/ul [CI = 0.22–0.35], healthy controls: median 0.28 [CI = 0.24–0.33]). We show that the CSF elevation of C4A in FEP-SCZ exceeds what can be expected from genetic risk variance in the *C4* locus, and in patient-derived cellular models we identify a mechanism dependent on the disease-associated cytokines interleukin (IL)−1beta and IL-6 to selectively increase neuronal *C4A* mRNA expression. In patient-derived CSF, we confirm that IL-1beta correlates with C4A controlled for genetically predicted *C4A* RNA expression ($r = 0.39$; CI: 0.01–0.68). These results suggest a role of C4A in early schizophrenia pathophysiology.

Schizophrenia (SCZ) is a highly heritable and polygenic brain disorder with the strongest associated SCZ locus located close to the complement component 4 (*C4*) genes. Distinct from mouse, human *C4* is encoded by two closely related genes, *C4A* and *C4B*, then typically present in multiple copy numbers (CNs) per genome and gene isotype. SCZ risk attributed to this locus can largely be explained by genetically predicted *C4A* RNA expression, and elevated *C4A* expression has been confirmed in SCZ postmortem brain tissue[1]. During brain development, microglia utilize complement signaling for selective removal of supernumerary synapses by complement receptor 3 (C3R)-dependent phagocytosis[2,3]. In line with the observed decrease in synapse density

in SCZ[4,5], excessive microglial synapse elimination has been observed in SCZ-derived in vitro models[6]. *C4A* CNs also correlate with complement deposition in patient-derived models, as well as complement-dependent microglial engulfment of synaptic structures[6]. Notably, *C4B* CNs, not linked to SCZ risk[1], do not influence neuronal complement deposition or synapse elimination in these models[6]. More recently, these in vitro findings were also confirmed in vivo using mouse models overexpressing mouse *C4*[7] or human *C4A*[8].

Despite the accumulating evidence from experimental models linking *C4A* to excessive synapse removal in SCZ, data showing increased in vivo protein levels in patients has been lacking. One

**a.**

**b.** **c.**

**d.** **e.**

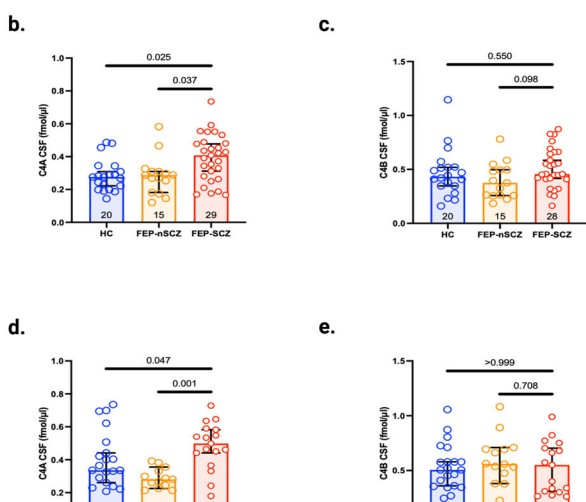

**Fig. 1 | Cerebrospinal fluid levels of C4A are increased in patients with first-episode psychosis who develop schizophrenia. a** Overview of the study design. **b** In the discovery cohort (KaSP), patients with first-episode psychosis (FEP) who developed schizophrenia (FEP-SCZ; $n = 29$) displayed significantly higher cerebrospinal fluid (CSF) C4A concentrations as compared to healthy controls (HCs; $n = 20$) or patients with FEP who did not develop SCZ (FEP-nSCZ; $n = 15$) (HCs: 0.28 fmol/ul; 95% confidence interval [CI] = 0.24–0.33, FEP-nSCZ: 0.29 fmol/ul; CI = 0.22–0.35, FEP-SCZ: 0.41; CI = 0.34–0.45, adjusted $P$ [FEP-SCZ vs. HCs]=0.025, adjusted $P$ [FEP-SCZ vs. FEP-nSCZ]=0.037). **c** CSF C4B concentrations were similar across groups (HCs [$n = 20$]: 0.43 fmol/ul; CI = 0.36–0.57, FEP-nSCZ [$n = 15$]: 0.38 fmol/ul; CI = 0.30–0.49, FEP-SCZ [$n = 28$]: 0.46 fmol/ul; CI = 0.44–0.58, adjusted $P$ [FEP-SCZ vs. HCs]=0.550, adjusted $P$ [FEP-SCZ vs. FEP-nSCZ]=0.098). **d** In the

replication cohort (GRIP), patients with FEP-SCZ ($n = 17$) displayed significantly higher CSF C4A concentrations as compared to HCs ($n = 21$) or patients with FEP-nSCZ ($n = 13$) (HC: 0.34 fmol/ul; CI: 0.31–0.47, FEP-nSCZ: 0.28 fmol/ul; CI: 0.25–0.33, FEP-SCZ: 0.50 fmol/ul; CI: 0.41–0.56, adjusted $P$ [FEP-SCZ vs. HCs] = 0.047, adjusted $P$ [FEP-SCZ vs. FEP-nSCZ]=0.001), while (**e**) CSF C4B concentration were similar across groups (HCs [$n = 21$]: 0.51 fmol/ul; CI = 0.44–0.63, FEP-nSCZ [$n = 14$]: 0.56 fmol/ul; CI = 0.47–0.72, FEP-SCZ [$n = 17$]: 0.55 fmol/ul; CI = 0.41–0.65, adjusted $P$ [FEP-SCZ vs. HCs]=0.999, adjusted $P$ [FEP-SCZ vs. FEP-nSCZ]=0.708). Bar graphs represent medians and error bars represent 95% CIs. Data were analyzed using Kruskal–Wallis $H$ tests followed by post-hoc tests. Significance was set to $P < 0.05$. All reported $p$-values are two-sided. Figure 1a was created with BioRender.com. Source data for graphs in Fig. **1b**–**e** are provided in the Source Data file.

reason is that these measurements have been limited by difficulties in distinguishing C4A and C4B as the peptide sequence only differ by a few amino acids. In the present study, we developed a targeted mass spectrometry method capable of detecting unique peptide sequences in C4A and C4B protein. We applied this method to human cerebrospinal fluid (CSF), collected from two independent cohorts of patients with first-episode psychosis (FEP), hypothesizing that patients with FEP who develop SCZ (FEP-SCZ) would display elevated CSF concentrations of C4A, although not C4B, as compared to matched healthy controls (HCs) (Fig. 1a). Taking advantage of the heterogeneity in clinical outcomes among patients with FEP[9], we also analyzed CSF from patients with FEP within the same cohorts that did not present with a SCZ diagnosis at follow-up (FEP-nSCZ). As polygenic risk scores have been shown to discriminate FEP-SCZ cases from FEP-nSCZ cases[10], we hypothesized that patients with FEP-SCZ would also display elevated CSF C4A concentrations as compared to patients with FEP-nSCZ, given that this within patients comparison would be less prone to confounding related to patient status.

## Results

### Patients with schizophrenia display elevated CSF C4A concentrations

In the first cohort (Karolinska Schizophrenia Project; KaSP) we included 44 patients with FEP and 20 age- and sex-matched HCs. Patients with FEP were stratified depending on if they subsequently developed SCZ (FEP-SCZ; $n = 29$) or not (FEP-nSCZ; $n = 15$). The three groups (HCs and two patient groups) displayed no significant differences in terms of demographics and clinical characteristics (Supplementary Table 1). We observed significantly higher CSF C4A concentrations in patients with FEP-SCZ as compared to HCs or patients with FEP-nSCZ (HCs: 0.28 fmol/ul; 95% confidence interval [CI] = 0.24–0.33, FEP-nSCZ: 0.29 fmol/ul; CI = 0.22–0.35, FEP-SCZ: 0.41; CI = 0.34–0.45, adjusted $P$ (FEP-SCZ vs. HCs) = 0.025, adjusted $P$ (FEP-SCZ vs. FEP-nSCZ) = 0.037);

Fig. 1b). On the contrary, CSF C4B concentrations were more similar across groups (HCs: 0.43 fmol/ul; CI = 0.36–0.57, FEP-nSCZ: 0.38 fmol/ul; CI = 0.30–0.49, FEP-SCZ: 0.46 fmol/ul; CI = 0.44–0.58, adjusted $P$ (FEP-SCZ vs. HCs) = 0.550, adjusted $P$ (FEP-SCZ vs. FEP-nSCZ) = 0.098; Fig. 1c). Fourteen patients with FEP-SCZ and eight patients with FEP-nSCZ patients had been prescribed an antipsychotic, although in no instance for more than one month before CSF collection. At group level, C4A concentrations were similar between medicated (an antipsychotic agent) and non-medicated patients (median: 0.34 fmol/ul [CI: 0.31–0.43]; median: 0.36 fmol/ul [CI: 0.32–0.43], respectively, $P = 0.931$), then suggesting that the elevation in C4A levels in patients with FEP-SCZ were not due to confounding by antipsychotic drugs.

### Replication of the initial findings from the discovery cohort

To replicate our initial finding, we used an independent FEP cohort (The Gothenburg research and investigation on psychosis project; GRIP), comprising of 31 patients with FEP (FEP-SCZ; $n = 17$, and FEP-nSCZ; $n = 14$) and 21 HCs. The FEP-nSCZ group included more smokers (64%) than the HC group (34%) but smoking was unrelated to CSF C4A or C4B concentration ($r = -0.15$; CI: $-0.42$–$0.15$ $P = 0.308$, and $r = -0.22$, CI: $-0.47$–$0.07$; $P = 0.124$, respectively). Otherwise, the three groups did not display any major differences in terms of demographics and clinical characteristics (Supplementary Table 2). In accordance with our results from the discovery cohort (KaSP cohort), we observed significantly higher CSF C4A concentrations in the FEP-SCZ group as compared to HCs or the FEP-nSCZ group (HC: 0.34 fmol/ul; CI: 0.31–0.47, FEP-nSCZ: 0.28 fmol/ul; CI: 0.25–0.33, FEP-SCZ: 0.50 fmol/ul; CI: 0.41–0.56, adjusted $P$ (FEP–SCZ vs. HCs)=0.047, adjusted $P$ (FEP–SCZ vs. FEP-nSCZ)=0.001); Fig. 1d), while CSF C4B concentrations were again more similar across groups (HC: 0.51 fmol/ul; CI: 0.44–0.63, FEP-nSCZ: 0.56 fmol/ul; CI: 0.47–0.72, FEP-SCZ: 0.55 fmol/ul; CI: 0.41–0.65, adjusted $P$ (FEP-SCZ vs. HCs)=0.999, adjusted $P$ (FEP-SCZ vs. FEP-nSCZ)=0.708; Fig. 1e. As in the discovery cohort, C4A

**Table 1 | Correlations of CSF C4A and C4B concentration to the Positive and Negative Syndrome Scale (PANSS) scores, and performance in the key domains of the MATRICS Consensus Cognitive Battery (MCCB)**

| | | C4A | | C4B | |
|---|---|---|---|---|---|
| | | FEP-nSCZ | FEP-SCZ | FEP-nSCZ | FEP-SCZ |
| *Positive and Negative Symptom Scale; PANSS* | | | | | |
| Positive sub-scale | r | −0.035 | −0.27 | 0.05 | 0.19 |
| | P | 0.906 | 0.136 | 0.863 | 0.435 |
| | N | 15 | 29 | 15 | 28 |
| Negative sub-scale | r | 0.10 | −0.25 | −0.23 | 0,00083 |
| | P | 0.906 | 0.218 | 0.567 | 0.997 |
| | N | 15 | 29 | 15 | 28 |
| General sub-scale | r | 0.04 | −0.02 | −0.52 | 0.29 |
| | P | 0.906 | 0.997 | 0.168 | 0.435 |
| | N | 15 | 29 | 15 | 28 |
| *MATRICS Consensus Cognitive Battery (MCCB)* | | | | | |
| TMT | r | −0.45 | 0.05 | 0.01 | −0.14 |
| | P | 0.32 | 0.817 | 0.992 | 0.971 |
| | N | 15 | 27 | 15 | 26 |
| BACS-SC | r | 0.19 | 0.15 | 0.01 | 0.0034 |
| | P | 0.713 | 0.778 | 0.988 | 0.987 |
| | N | 15 | 27 | 15 | 26 |
| HVLT-R | r | 0.30 | 0.34 | 0.988 | 0.05 |
| | P | 0.566 | 0.433 | 0.776 | 0.971 |
| | N | 15 | 25 | 15 | 26 |
| WMS III SS | r | 0.47 | 0.10 | −0.06 | 0.07 |
| | P | 0.320 | 0.798 | 0.988 | 0.971 |
| | N | 15 | 27 | 15 | 26 |
| LNS | r | 0.01 | −0.17 | 0.04 | −0.06 |
| | P | 0.977 | 0.751 | 0.776 | 0.971 |
| | N | 15 | 27 | 15 | 26 |
| NAB-MAZES | r | 0.33 | 0.03 | 0.18 | −0.03 |
| | P | 0.566 | 0.987 | 0.988 | 0.971 |
| | N | 15 | 27 | 15 | 26 |
| BVMT-R | r | 0.47 | 0.29 | −0.19 | −0.07 |
| | P | 0.320 | 0.496 | 0.988 | 0.971 |
| | N | 15 | 27 | 15 | 26 |
| Fluency | r | −0.15 | 0. | −0.36 | 0.13 |
| | P | 0.733 | 0.978 | 0.988 | 0.971 |
| | N | 15 | 27 | 15 | 26 |
| MSCEIT ME | r | −0.26 | 0.25 | 0.23 | −0.16 |
| | P | 0.583 | 0.729 | 0.988 | 0.971 |
| | N | 15 | 27 | 15 | 26 |
| CPT-IT | r | 0.11 | 0.29 | 0.05 | −0.05 |
| | P | 0.783 | 0.164 | 0.988 | 0.971 |
| | N | 15 | 27 | 15 | 25 |

Data were generated from a total of 44 patients in the KaSP cohort with available data (median age = 24; interquartile range: 22–33; 16 females and 28 males). Number of included subjects are given in the table for each analysis. Correlations were performed using Spearman correlation analyses (two-sided *P* values corrected for multiple testing using Benjamini-Hochberg false discovery rate correction).
*FEP-nSCZ* first-episode patients that did not develop schizophrenia, *FEP-SCZ* first-episode patients that developed schizophrenia

concentrations in the antipsychotic exposed and the non-antipsychotic exposed patients were not significantly different (median: 0.39 fmol/ul; CI: 0.35–0.50, and median: 0.33 fmol/ul; CI: 0.27–0.41, respectively, *P* = 0.193).

Combining both cohorts (KaSP and GRIP), we observed elevated CSF C4A concentrations in patients with FEP-SCZ as compared to both patients with FEP-nSCZ (57% increase) and HCs (51% increase), while CSF C4B concentrations were not significantly different between the groups (Supplementary Fig. 1). To exclude potential residual confounding by age, sex, or smoking, we also performed an analysis adjusting for these factors and with similar results (Supplementary Fig. 1). In the combined sample, a total of 31 patients with FEP were not on an antipsychotic agent (12 FEP-nSCZ patients and 19 FEP-SCZ patients), while 43 patients had a short-term exposure for an antipsychotic agent (16 FEP-nSCZ patients and 27 FEP-SCZ patients). In both groups (i.e., medicated, and non-medicated), we observed a similar and significant increase of CSF C4A concentration in patients with FEP-SCZ as compared to patients with FEP-nSCZ patients (Supplementary Fig. 2).

### CSF C4A concentrations associate with habituation of the startle response

Clinical data regarding severity and symptom profiles were available for the KaSP cohort (*n* = 44) and were used for exploratory analyses of correlations between CSF C4A concentrations and disease phenotypes. After controlling for multiple testing, we observed no significant correlations between CSF C4A concentration and scores on the positive, negative, and general psychopathology scale of the Positive and Negative Syndrome Scale (PANSS), or performance in the key domains of the MATRICS Consensus Cognitive Battery (MCCB) (Table 1). We then studied CSF C4A in the context of SCZ-related impairment of sensorimotor gating by measuring pre-pulse inhibition (PPI) and habituation due to repeated exposure to a stimulus[11,12]. Similar to findings in mouse[8], CSF C4A displayed no significant association with % PPI at either interstimulus interval (Supplementary Table 3). However, instead we observed significant inverse correlations between CSF C4A concentration and habituation (habituation 2: *r* = 0.37, CI: 0.07–0.61; *P*(adjusted) = 0.036, habituation 3: *r* = 0.43, CI: 0.14–0.65 *P*(adjusted) =0.036; habituation 5: *r* = 0.35, CI: 0.05–0.60 *P*(adjusted)=0.040, habituation 6: *r* = 0.39, CI: 0.09–0.63 *P*(adjusted)=0.036, habituation 9: *r* = 0.37, CI:0.07–0.61 *P*(adjusted)=0.036, habituation 10: *r* = 0.39, CI: 0.09–0.62 *P*(adjusted)=0.036, and habituation 14: *r* = 0.37, CI: 0.07–0.60 *P*(adjusted)=0.036; Supplementary Table 3), a deficit hypothesized to reflect impairment in synaptic plasticity[13].

### Elevation of CSF C4A exceeds genetically predicted *C4A* RNA expression

Despite a strong correlation between *C4A* CNs and *C4A* RNA expression in the brain[1], previous studies suggest that upregulation of *C4A* expression in SCZ exceeds what can be expected from genetic risk variance in the *C4* locus[14]. To test this at the protein level, we used droplet digital PCR (ddPCR) to measure *C4A* CNs and *HERV* insertions in patients with FEP that had available DNA and CSF C4A protein levels in the GRIP cohort (*n* = 22 FEP and *n* = 21 HCs). We observed a moderate correlation between *C4A* CNs and CSF C4A protein levels (*r* = 0.48; CI: 0.21–0.68; *P* = 0.001), as well as between genetically predicted *C4A* RNA expression[1] and CSF C4A protein levels (*r* = 0.44; CI: 0.16–0.65; *P* = 0.003; Supplementary Fig. 3). The correlation between *C4B* CNs and CSF C4B protein levels was weaker and did not reach significance in this sample (*r* = 0.19; CI: −0.11–0.46; *P* = 0.205; Supplementary Fig. 3). As a sensitivity analysis, we then re-measured CSF concentrations of another C4B peptide and observed a similar and non-significant correlation to *C4B* CNs as in the original analysis (*r* = 0.22; CI: −0.08–0.49; *P* = 0.146; Supplementary Fig. 3). This suggests that CSF C4A protein levels are under a stricter genetic control from the *C4* locus than CSF C4B protein levels. With available *C4A* CNs, as measured by ddPCR, we then adjusted the original analysis, comparing CSF C4A concentration across groups, for *C4A* CNs. CSF C4A

protein levels per *C4A* CN (as well as predicted *C4A* RNA expression) then remained significantly elevated in patients with FEP-SCZ as compared to patients with FEP-nSCZ and HCs (HC: HCs: $n = 21$ median = 0.17, CI: 0.15–0.21 and median = 0.32, CI: 0.27–0.38; respectively, FEP-nSCZ: $n = 10$ median = 0.16, CI: 0.13–0.20 and median = 0.29, CI: 0.24–0.33; respectively, FEP-SCZ: $n = 12$ median = 0.23, CI: 0.19–0.28 and median = 0.46, CI: 0.35–0.56; respectively, $P$ (FEP-SCZ vs. HCs) =0.039 and 0.031 respectively, $P$ (FEP-SCZ vs. FEP-nSCZ)=0.029 and 0.020 respectively; Supplementary Fig. 4), then confirming that factors independent of genetic risk variance within the *C4* locus contributes to the increase in CSF C4A protein levels in SCZ. However, CSF C4B protein levels were not significantly different when adjusting for copy numbers (as well as predicted *C4B* RNA expression) between FEP-SCZ and HC or FEP-nSCZ (Supplementary Fig. 4).

### Disease-associated cytokines increases *C4A* mRNA expression in patient-derived cellular models

Recent brain co-expression network analyses[14], using *C4A* as a seed gene, have revealed an enrichment of genes involved in immune processes, such as NF-κB signaling, rather than other complement system genes, and experimental studies suggest that cytokines can modulate the expression of the *C4* genes, although in a tissue-dependent fashion and with a variable effect on *C4A* versus *C4B* expression[15,16]. The pro-inflammatory cytokines, interleukin(IL)−1beta and IL-6 both have repeatedly been shown to be elevated in CSF of patients with SCZ[17,18], and we therefore explored their influence on neuronal *C4A* expression. To this end, we generated stably inducible *neurogenin 2* (*NGN2*) neural progenitor cells from induced pluripotent stem cell (iPSC) of four patients and differentiated these cells to cortical excitatory neurons. These derived cultures were then stimulated for 24 hours with either IL-1beta (1 ng/ml), IL-6 (1 ng/ml), or a combination of IL-1beta and IL-6 (Fig. 2a). We observed significantly increased neuronal mRNA expression of *C4A* as measured by qPCR in cultures stimulated with a combination of IL1beta and IL-6 as compared to vehicle (VEH) stimulated cultures from the same patients (Fig. 2b), while no significant effect was evident on the relative *C4B* mRNA expression (Fig. 2c), then suggesting that the IL1beta/IL-6 combination foremost induces *C4A* mRNA expression. However, the relative increase in mRNA expression by IL-1 beta/IL-6 induction could theoretically vary by CNs for the corresponding *C4* gene. Thus, we specifically looked at the data from one of the patient-derived line (across three independent experiments) that had been genotyped for *C4* haplotypes (ddPCR) and had equal CNs of *C4A* and *C4B* (two CNs for each gene). The significant increase in relative *C4A* mRNA expression after IL1beta/IL-6 stimulation remained when only analyzing cultures from this line (VEH: median (fold change) = 1.0, CI:0.66-1.5, IL-1beta: median = 0.98, CI:0.50-1.7, IL-6: median = 2.1, CI:1.1-2.4, IL1beta/IL-6: median=2.5, CI:2.0-3.5, Kruskal-Wallis *H* test: adjusted $P$ (VEH vs. IL-1beta) = 0.999, adjusted $P$ (VEH vs. IL-6)=0.285, and adjusted $P$ (VEH vs. IL1beta/IL-6) = 0.004), while no significant increase was observed for *C4B* mRNA expression after IL-1beta/IL-6 stimulation (VEH: median = 1.0, CI:0.63-1.5, IL-1beta: median = 1.2, CI: 0.71–2.1, IL-6: median = 1.9, CI: 0.85–3.0, IL1beta/IL-6: median = 2.4, CI: 1.2–3.7, Kruskal–Wallis *H* test: adjusted $P$ (VEH vs. IL-1beta) = 0.999, adjusted $P$ (VEH vs. IL-6) = 0.411, and adjusted $P$ (VEH vs. IL1beta/IL-6) = 0.094).

### IL-1beta correlates with C4A in CSF of patients with schizophrenia

To corroborate this observation in a clinical context, and to determine the relative importance of each cytokine in clinically relevant concentrations, we measured CSF IL-1beta and IL-6 in 26 patients with FEP and available CSF (KaSP cohort; Fig. 3a). Controlling for genetically predicted *C4A* RNA expression using whole genome sequencing data[1], we observed a significant correlation between IL-1beta and CSF C4A concentration ($r = 0.39$; CI: 0.01–0.68; $P = 0.047$; Fig. 3b), while IL-6 did

not significantly influence C4A levels ($r = −0.047$; CI: −0.42–0.34; $P = 0.820$; Fig. 3c). In contrast, there were no significant correlations between these two cytokines and C4B concentrations in CSF ($r = 0.29$; CI: −0.11–0.61; $P = 0.150$; Fig. 3d, and $r = 0.32$ CI: −0.09–0.64; $P = 0.112$; Fig. 3e, respectively). This suggests that an IL-1beta/IL-6 dependent mechanism, detected in our in vitro model and predominately influencing *C4A* mRNA expression, is also relevant in an in vivo context, with variance in vivo CSF levels of IL-1beta influencing C4A protein levels independent of *C4A* CNs (see Supplementary Fig. 5 for a comparison of the elevation of CSF IL-1beta and C4A in patients with FEP-SCZ as compared to patients to FEP-SCZ).

### C4A correlates inversely with measurements of synapse density in CSF

As *C4A* CNs predict synaptic complement deposition and synaptic pruning in experimental patient-derived models[6], we explored associations between CSF C4A protein concentration and in vivo proxies of synapse density. CSF levels of neuronal pentraxins (1, 2, and the receptor) were recently shown to inversely correlate with prefrontal cortical thickness determined by Magnetic Resonance Imaging (MRI), as well as to predict synapse loss in dementia and Alzheimer's disease[19,20]. Thus, we measured these three markers in CSF from participants in the KaSP cohort (patients with FEP: $n = 43$, HCs: $n = 19$). After adjusting for age, sex, and diagnosis, we observed significant negative correlations in CSF between C4A and neuronal pentraxin 1 ($r = −0.31$; $P$(adjusted) = 0.026), neuronal pentraxin 2 ($r = −0.27$; $P$(adjusted) = 0.026), and the neuronal pentraxin receptor ($r = −0.34$; $P$(adjusted) = 0.040), see Fig. 4a–c.

## Discussion

This study demonstrates that C4A protein concentrations are specifically elevated in the CSF of patients with FEP who subsequently develop SCZ, strengthening the notion of a role for this complement component in the early pathophysiological events of SCZ. Furthermore, in line with the effect of *C4A* CNs on microglial synapse engulfment in patient-derived models[6], we show that C4A protein levels in patient-derived CSF inversely correlate with previously validated proxies of synapse density. This emphasizes the in vivo relevance of C4A for synapse density in SCZ and supports the notion from human experimental studies that *C4A* CNs influence SCZ risk by causing excessive synaptic pruning. Although the elevation of CSF C4A protein levels in patients with FEP who develop SCZ was modest, it was still larger than expected from previous genetic studies. Thus, on protein level we corroborate previous findings from gene expression analyses that *C4A* CNs do not fully explain the observed *C4A* RNA upregulation in SCZ[1,14]. Using a patient-derived cellular model, we show that IL-1beta/IL-6 induces neuronal *C4A* mRNA expression. As patients with SCZ as a group display elevated levels of these cytokines in CSF[17,21], the increase in C4A levels per *C4A* CNs can therefore be expected to be more pronounced in SCZ as compared to HCs and partly explain the remaining elevation in patients with FEP-SCZ after adjusting for *C4A* CNs. This hypothesis is supported by our CSF analyses, where we show that IL-1beta levels correlate with CSF C4A levels per *C4A* CNs, suggesting in vivo relevance. The results are in line with studies showing that immune activation in SCZ, including increased levels of pro-inflammatory cytokines such as IL-1beta and IL-6, are linked to a more pronounced cortical thinning and worsen cognitive performance[22–24]. Further, IL-1beta has been proposed as a biomarker also for several neurocognitive phenotypes beyond SCZ[25,26], as well as in suicidal behavior[27,28]. Further studies are warranted to evaluate if a mediation by induction of *C4A* mRNA expression and excessive synaptic pruning contributes to these phenotypes.

Utilizing detailed clinical data regarding severity and symptom profiles we also study within-subject correlations to CSF C4A levels. We observe a clear inverse correlation between CSF C4A concentration

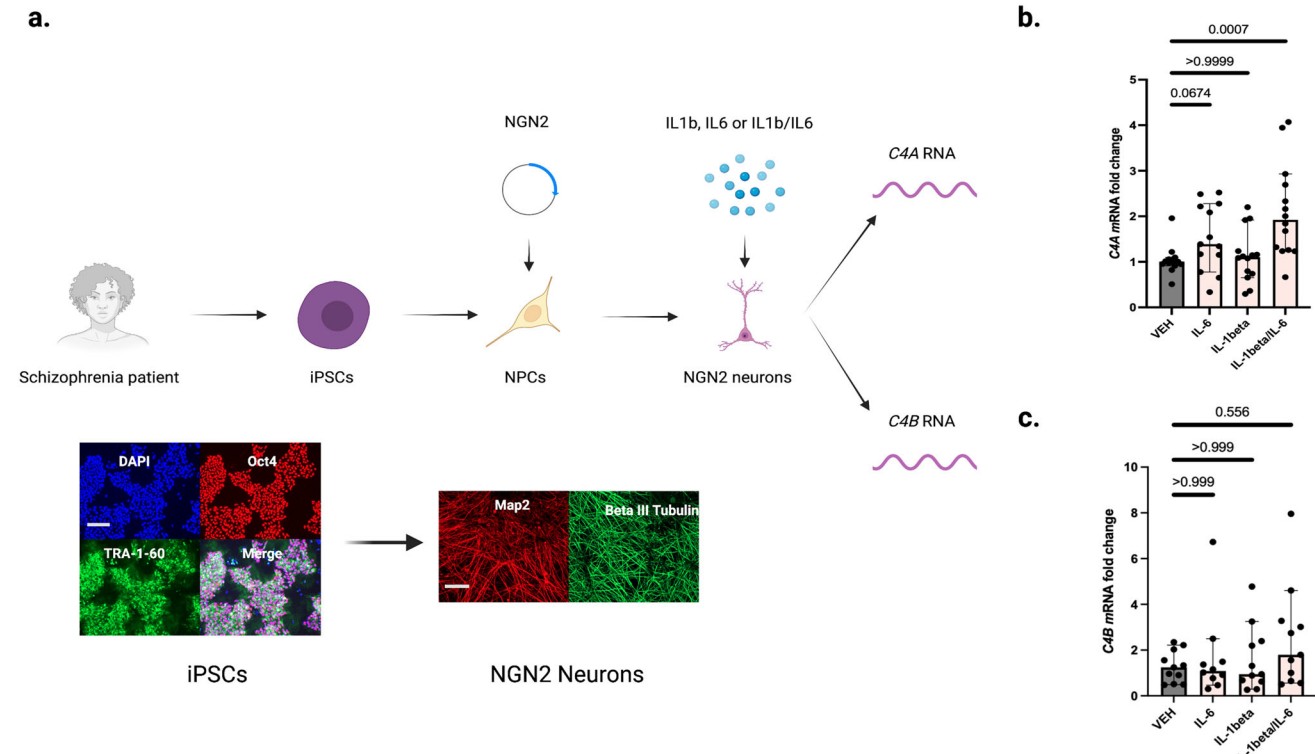

**Fig. 2 | Disease-associated cytokines increase neuronal *C4A* mRNA expression in a patient derived schizophrenia model. a** Overview of in vitro experiments in which induced pluripotent stem cells (iPSCs) derived from four chronic patients (all white males, 28- years old, 44-years old, 55-years old, and 58-years old, respectively) were differentiated into cortical excitatory neurons. Representative quality control immunocytochemistry images display octamer binding transcription factor 4 (POU domain, class 5, transcription factor 1) and TRA-1–60, as well as nuclear staining with DAPI for iPSCs that were used to derive cortical excitatory neurons (here stained for MAP2 and Beta III Tubulin). Scale bars for representative images:100 uM. **b** qPCR was used to measure relative mRNA expression of *C4A*. The relative *C4A* mRNA expression increased significantly after pre-treatment for 24 h with the interleukin (IL) −1beta and IL-6 combination (median fold change = 1.9, 95% confidence interval [CI] = 1.5–2.7) compared to vehicle (VEH) stimulated neuronal cultures (median = 1.0, CI = 0.86–1.2, adjusted $P$ = 0.0007), while the increase in relative mRNA expression was not significant after IL-1beta (median = 1.1, CI = 0.79–1.5, adjusted $P$ = 0.999) or IL-6 stimulation (median = 1.4, CI = 1.1–2.0 adjusted $P$ = 0.067). $n$ = 14 biologically independent derivations (datapoints) per condition ($n$ = 13 for IL-6 exposure condition). qPCR reactions were done in duplicate and averaged. The experiment was repeated three times. As a sensitivity analysis, we also analyzed the data using a generalized linear mixed effect model (robust estimation to handle violations of model assumption and adjusting for multiple testing using sequential Sidak). Target:

CSF C4A concentration, fixed effects: experimental condition (VEH, IL-1beta, IL-6, IL1-beta/IL-6), cell donor ($n$ = 4 categories), and experimental round ($n$ = 3 categories). Significant effects were observed for experimental condition ($F$ = 8.9; $P$ = 2.2 × 10⁻⁴; estimated means: VEH = 0.78 [CI = 0.44–1.01], IL1-beta=0.78 [CI = 0.39–1.17], IL-6 = 1.23 [CI = 0.84–1.62], IL-1beta/IL-6 = 1.78 [CI = 1.36–2.20], fixed effect coefficient for VEH condition [IL-1beta/IL-6 as reference group] = −1.06 [CI = −1.53 to −0.58]; $P$ = 4.6 × 10⁻⁵) and for donor ($F$ = 4.6; $P$ = 0.007). **c** Compared to the vehicle (VEH) stimulated neurons (median fold change = 1.2, CI = 0.82–1.7) the relative *C4B* mRNA expression was not significantly increased after IL-1beta stimulation (median fold change = 0.94, CI = 0.65–2.6, adjusted $P$ = 0.999), IL-6 stimulation (median = 1.1, CI = 0.33–3.0, adjusted $P$ = 0.999), or IL-1beta/IL-6 stimulation (median fold change = 1.8, CI = 1.0–4.0, adjusted $P$ = 0.556). $n$ = 11 biologically independent derivations (datapoints) per condition ($n$ = 10 for IL-6 exposure condition). qPCR reactions were done in duplicate and averaged. ΔCt values were obtained by comparison to Ct values of a housekeeping gene, and fold changes (2 − ΔΔCt) were then generated by normalization to the average ΔCt value for VEH treated wells per plate (see Methods). Similar data were observed across three independent experiments. Kruskal−Wallis $H$ tests followed by post-hoc tests. Significance was set to $P$ < 0.05. All reported $p$-values are two-sided. Figure 2a was created with BioRender.com. Source data for graphs in Fig. 2b, c are provided in the Source Data file.

and habituation of the startle response. Patients with SCZ as a group display decreased habituation of the startle response[29], and from a mechanistic perspective habituation largely relies on synaptic plasticity. Thus, our data suggest that patients with SCZ and a high *C4A* CN burden may exhibit more difficulties with habituation due to inappropriate neuronal wiring. Notably, we observe no significant correlations with performance measurements across the domains in the MATRICS test battery. This is at odds with previous studies reporting inverse associations between genetically predicted *C4A* expression (imputed from common variants) and performance in tests measuring memory aspects and processing speed[30,31]. While it is likely to assume that the use of CSF protein levels reduced non-differential bias, the obvious reduction in sample size (while maintaining a large number of tested variables, i.e., need for multiple testing correction) made our sample less suitable for studying the influence of the *C4* locus on cognition due to limited power.

Several previous studies have measured total C4 levels in plasma from patients with SCZ but with conflicting results[32–35], and C4 levels in plasma do not significantly correlate with total C4 levels in CSF[36]. In the current study, we separate C4A and C4B levels in CSF and in line with previous experimental and genetic studies we observe that only C4A is increased in first episode patients with SCZ. A recent study reported elevated total C4 concentrations in 32 patients with chronic SCZ spectrum disorder (median duration of illness of 14 years) with ongoing antipsychotic medication[36]. Thus, we also measured a peptide corresponding to total C4 levels in CSF but unlike the previous study of chronic patients with schizophrenia we did not observe a significant elevation of total C4 levels in CSF of patients with FEP (Supplementary Fig. 6). Similar to this study, however, we observed an age effect on CSF total C4 levels (Supplementary Fig. 7), but by separating CSF C4A and C4B levels, we could show that this effect was driven by an age-dependent increase in C4B levels (Supplementary Fig. 8), with C4B

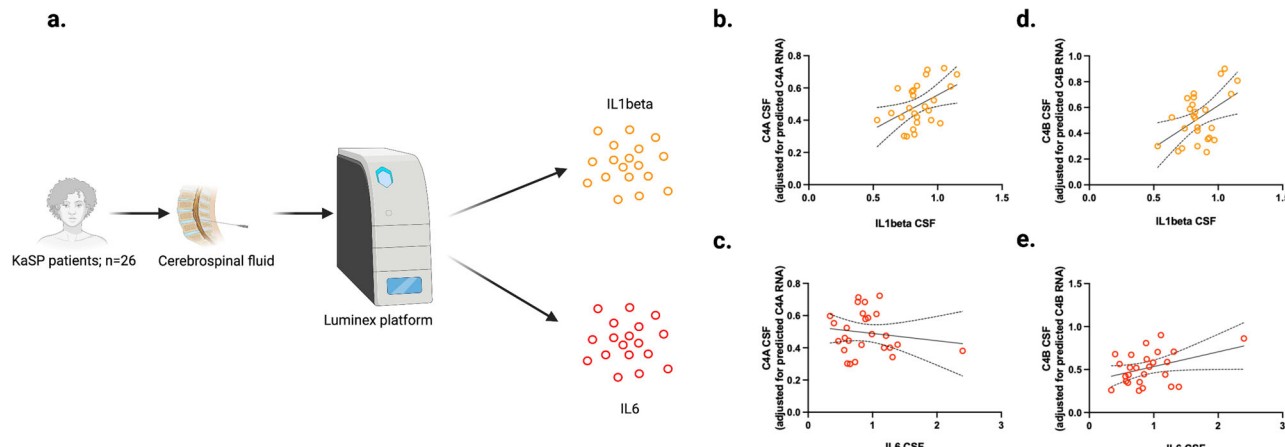

**Fig. 3 | Disease-associated cytokines associate with C4A protein levels in cerebrospinal fluid of patients with first-episode psychosis. a** Overview of cytokine analyses. Cerebrospinal fluid (CSF) from patients with first-episode psychosis in the KaSP cohort (*n* = 26) were analyzed for interleukin (IL)−1beta, IL-6, and C4A levels. Median age in the cohort was 29 years (interquartile range: 23-32) and 18 participants were males. **b** CSF C4A protein levels (adjusted for genetically predicted *C4A* RNA expression) correlated with IL-1beta levels (*r* = 0.39; 95% confidence interval [CI]: 0.01–0.68, *P* = 0.046), **c** but not IL-6 levels (*r* = −0.05 CI: −0.44–0.36, *P* = 0.820). **c** C4B protein levels (adjusted for genetically predicted *C4B* RNA expression) did not correlate significantly with IL-1beta levels (*r* = 0.29 CI: −0.12–0.62, *P* = 0.150; **c**), or **d** IL6 levels (*r* = 0.32 CI: −0.09–0.64, *P* = 0.112). Data were analyzed by creating CSF C4A (or C4B) levels adjusted for predicted RNA expression (by extracting unstandardized residuals from a linear regression model analyzing predicted RNA expression vs. measured CSF protein levels), and then used in Spearman correlation analyses against CSF cytokine levels. All reported *P* values are two sided. Intercepts and 95% CI bands are indicated in figure panel 3**b**–**d**. Source data for graphs in figure panels 3**b**–**e** are provided in the Source Data file. Figure 3a was created with BioRender.com.

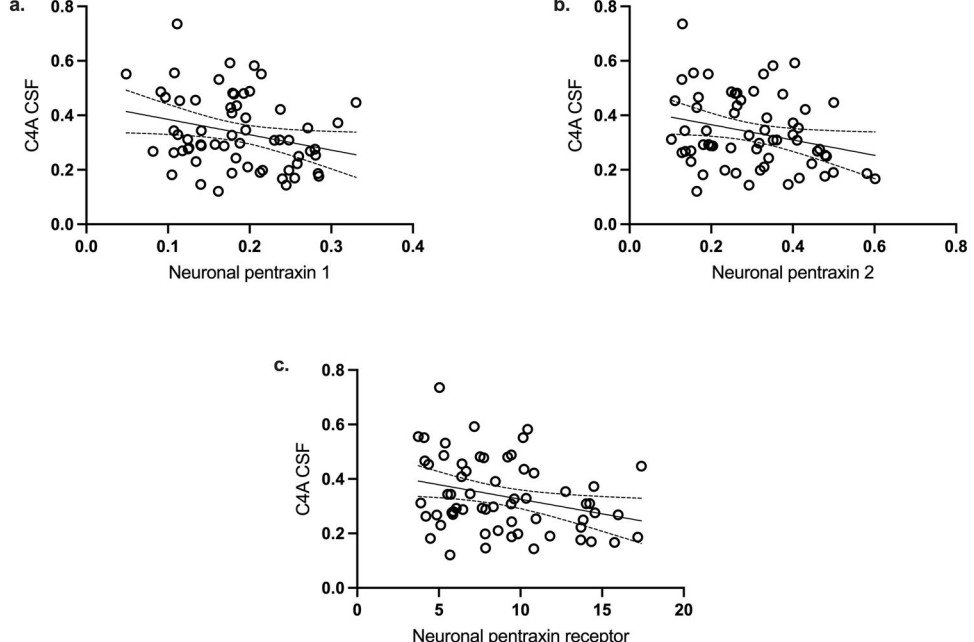

**Fig. 4 | Correlations of cerebrospinal fluid C4A and C4B protein concentration with markers of synapse density.** Cerebrospinal fluid (CSF) from 19 healthy controls, 14 patients with first-episode psychosis (FEP) who developed schizophrenia and 29 patients with first-episode psychosis (FEP) who developed schizophrenia in the KaSP cohort were analyzed. Median age: 27 years (interquartile range 22–32), 40 males and 22 females. CSF C4A levels (adjusted for age, sex, and diagnosis) were negatively correlated to CSF levels of neuronal pentraxin 1 [*r*(adjusted) = −0.30, *P*(adjusted) = 0.026; **a**], neuronal pentraxin 2 [*r*(adjusted) = −0.26; *P*(adjusted) = 0.026; **b**], and neuronal pentraxin receptor [*r* = −0.34; *P*(adjusted) = 0.040; **c**]. All correlations were analyzed using Spearman's partial correlation analyses. All reported *P* values are two sided and adjusted for multiple testing using Benjamini-Hochberg correction. Intercepts and 95% confidence intervals are indicated in the graphs. Source data for the graphs are provided in the Source Data file.

levels per *C4B* CNs being significantly higher than C4A levels per *C4A* CNs in human CSF (Supplementary Fig. 9).

In conclusion, this study shows that SCZ risk variants that increase *C4A* RNA expression also led to an increase in C4A protein levels in CSF and a concomitant decrease in proxies for synapse density. In two independent cohorts, we observe higher CSF C4A concentrations in

patients with FEP who go on to develop SCZ compared with those who get another diagnosis as well as in comparison to controls, and this also pertains to patients that were not on an antipsychotic agent. We show that this increase cannot be fully explained by genetic risk variants in the *C4* locus, and in patient-derived models we discover a mechanism by which the SCZ-associated cytokines IL-1beta and IL-6, previously

shown to be increased in SCZ, can selectively induce *C4A* mRNA expression. We then corroborate these results in patient-derived CSF analyses and show that IL-1beta correlates with C4A levels independent of *C4A* CNs. Taken together, these results confirm a role of C4A in early SCZ pathophysiology and suggest a targetable mechanism that can contribute to the increase in C4A levels and excessive synapse elimination.

## Methods

### Ethical statement

The study was approved by the Regional Ethics Committee in Stockholm (Sweden) and the Institutional Review Board of Partners HealthCare (MA, USA). Informed consent was obtained from all included subjects (CSF sampling and skin biopsies for deriving iPSCs).

### Study population and diagnostic assessments

KaSP Cohort: Karolinska Schizophrenia Project (KaSP) is a Swedish multidisciplinary research consortium investigating the pathophysiology of FEP and SCZ. Patients who seek health care for psychotic symptoms for the first time are recruited at psychiatric emergency wards and in- or outpatient facilities at the psychiatric clinics located in Stockholm, Sweden. Exclusion criteria are ongoing or previous prescription of an antipsychotic for more than 30 days, severe somatic and neurological diseases, current substance abuse (except nicotine use), or autism spectrum disorder. These patients are excluded by clinical examination, medical history, routine laboratory tests, including screening for drugs and MRI scans. All KaSP study participants were white. A baseline diagnosis is established based on a structured clinical interview of the Diagnostic and Statistical Manual of Mental Disorders IV, DSM-IV (Structured Clinical Interview for DSMIV-Axis I Disorders, SCID-I) at the time of inclusion. For the subdivision of patients with FEP into a FEP-SCZ and a FEP-nSCZ group we used data from a 1.5-year follow-up (median time to follow-up: 1.6 years). The same diagnostic procedure as at baseline was used. Patients who did not receive a SCZ diagnosis at follow-up received a diagnosis of delusional disorder, unspecified psychosis, acute and transient psychotic disorder, schizoaffective disorder, or major depressive disorder.

HCs were recruited by advertisement and matched on age and sex. Eligibility was determined by medical history, routine laboratory tests, clinical examination, and an MRI examination, as evaluated by an experienced neuroradiologist at the MR Centre, Karolinska University Hospital, Solna. The Mini International Neuropsychiatric Interview (MINI), performed by either a resident or a specialist in psychiatry, was used to exclude previous or current psychiatric illness. Further exclusion criteria were former or current use of illegal drugs, first-degree relatives with psychotic illness or bipolar disorder, as well as neurologic disease and/or severe somatic disease. All participants (patients and controls) were free from any form of substance abuse evaluated with Alcohol Use Disorders Identification Test (AUDIT) and the Drug Use Disorders Identification Test (DUDIT) at the time of the study. All eligible subjects with available data were included in the present study.

GRIP cohort: The Gothenburg research and investigation on psychosis (GRIP) is a study collecting data from patients with schizophrenia spectrum disorders at the psychosis clinic, Sahlgrenska University Hospital, Mölndal, Sweden. The initial diagnostic assessments are performed by treating clinicians at outpatient tertiary-care units, as well as the inpatient units, in Sahlgrenska University Hospital (Sweden). Further clinical information is obtained from interviews at tertiary-care units and investigation units, and when needed supplemented with information from clinical records. To secure diagnoses for research, a series of case conferences are then held (for this study in between October 2017–March 2018). A minimum of two board-certified psychiatrists participates in these case conferences where a consensus diagnosis is established according to the Diagnostic and

Statistical Manual of Mental Disorders, Fifth Edition (DSM-5). The diagnosis is based on structured interviews, medical records (also collected post-baseline), and when needed supplemented with information from the treating psychiatrist. The participating research clinicians are blinded to the results of the CSF analyses. The median time from inclusion in the study (when the initial diagnosis by the treating physician is made) to the establishment of a research consensus diagnosis on the case conference was here 1.8 years. For patients with FEP, the median time from first contact with psychiatric care to referral for further investigation of psychotic symptoms at the tertiary unit (i.e., study baseline) was 2 years. This time period, as well as time to a SCZ diagnosis within the study period, was unrelated to CSF C4A (Spearman correlation: $r = 0.15$ CI: $-0.25-0.50$, $P = 0.454$) or C4B levels (Spearman correlation: $r = 0.26$ CI: $-0.13-0.58$, $P = 0.170$), as well as to a SCZ- vs. a non-SCZ-related diagnosis (Spearman correlation: $r = 0.06$ CI: $-0.32-0.43$, $P = 0.746$). Patients who did not receive a SCZ diagnosis at the diagnosis evaluation around 2 years after baseline received a diagnosis of bipolar disorder, brief psychotic disorder, substance-induced psychotic disorder (not known at the initial evaluation), or a psychotic disorder not otherwise specified.

As no matched HCs are recruited in GRIP, we used HCs from the larger "St. Göran projektet" research project in which GRIP is a sub-study focusing on psychosis. These HCs are randomly selected by Statistics Sweden and undergo the same clinical investigations as patients and do not fulfil the criteria for a DSM-IV-TR disorder. From this cohort, we then selected HCs to as closely as possible match the patients in GRIP dependent on sex and age. All eligible subjects with available data were included in the present study.

### Lumbar puncture and collection of cerebrospinal fluid

KaSP cohort: Study participants fasted overnight, lumbar punctures were performed between 7.45 am–15.15 pm, using a non-cutting (atraumatic) 22 G spinal needle inserted at the L4–5 interspace with collection of 18 mL of CSF. Not more than 1 h after collection, CSF was centrifuged at 3500 rpm for 10 min divided into 10 aliquots and stored at −80 °C until analysis. All CSF samples were individually analyzed.

GRIP cohort: Subjects fasted overnight, and the lumbar puncture was performed between 10.45 and 11.15 am. A non-cutting (atraumatic) 22 G spinal needle was inserted into the L4/L5 interspace and about 15 ml of CSF was collected, gently inverted to avoid gradient effects, centrifuged for 10 min at 3500 rpm at 20 degrees, not more than 30 min after sampling. CSF was then divided into 0.5 ml aliquots that were stored at −80 °C pending analysis. All CSF samples were individually analyzed.

### Targeted mass spectrometry (C4A and C4B)

The protein concentration of the CSF samples was measured in triplicates using Micro BCA Protein Assay Kit (Thermo Fisher Scientific Inc, USA PN: 23235) prior to sample preparation. The measurements were performed in a 96-well plate (VWR International AB, Sweden PN: 738-0170) and fluorescence read by plate reader FLUOstar Omega. (BMG LABTECH, Germany). Sample preparation was performed on the Agilent AssayMAP Bravo Platform. Fifty μg protein of each CSF sample was used for analysis. Depending on the protein concentration of the samples, the appropriate volume of each sample was transferred to a 96-well plate (Greiner G650201) and 200 mM triethylammonium bicarbonate buffer (TEAB, Thermo Fisher Scientific, USA PN: 90114) manually added to a final volume of 50 μL. The Bravo platform was used for the following steps; addition of 50 μL TEAB buffer and 50 μL 3.3% solution of sodium deoxycholate (SDC, Sigma-Aldrich, Sweden PN: L3771) for dissociation of proteins (final concentration of 1%). Proteins were reduced by the addition of 10 μL 80 mM tris(2-carboxyethyl) phosphine (TCEP, final concentration of 5 mM, SigmaAldrich, Sweden PN: C4706) and the reaction was carried out for 1 h at 55 °C. Alkylation was carried out by the addition of 15 μL 120 mM

iodoacetamide (final concentration of 10 mM, Sigma-Aldrich, Sweden PN: I1149) for 30 min in the dark at room temperature. One μg trypsin (Sequencing Grade Modified, Promega, Sweden PN: V5111) was added to each sample and incubated overnight at 37 °C. The CSF samples were divided into four groups for sample preparation, and they were labeled with #1, #2, #3, and #4. In group #3, the protein concentration was lower than in the other groups and the starting volume for each sample was therefore more than 50 μL. To these samples, 200 mM triethylammonium bicarbonate buffer was manually added to a total volume of 100 μL and additional liquid handling steps were carried out on the Bravo platform. The trypsin digestion was quenched by the addition of 30 μL 10% FA (Thermo Fisher Scientific, USA PN: 94318) and the SDC pellet was removed by filtering the samples through a polypropylene filter plate (Agilent Technologies, Sweden, PN: 200931-100) with hydrophilic PVDF membrane (mean pore size 0.45 μm). The tryptic peptide concentration was detected using the Pierce Quantitative Colorimetric Peptide Assay (Thermo Fisher Scientific, USA PN: 23275). Each sample was spiked with a mixture of heavy isotope-labeled peptide standards (AQUA QuantPro peptides from Thermo Fisher Scientific) before LC-MS analysis.

The LC-MS analysis was performed on a Tribrid mass spectrometer Fusion equipped with a Nanospray Flex ion source, coupled to an EASY-nLC 1000 ultra-high pressure liquid chromatography (UHPLC) system (Thermo Fisher Scientific, USA). The tryptic CSF peptides spiked with AQUA peptides, were loaded on to an Acclaim PepMap 100 C18 precolumn (75 μm × 2 cm, Thermo Fisher Scientific, USA) and separated on an Acclaim PepMap RSLC column (75 μm × 25 cm, nanoViper, C18, 2 μm, 100 Å) with the flow rate 300 nL/min. The column was kept at 40 °C using a Phoenix S&T Nano LC column heater (MS Wil GmbH, Zurich). Solvent A (0.1% FA, Sigma-Aldrich) and solvent B (0.1% FA in acetonitrile, SigmaAldrich) were used for the nonlinear gradient. The percentage of solvent B was 1% for the first 3 min and increased to 30% in 50 min and then to 90% in 5 min which was kept for 7 min to wash the column. The tryptic CSF peptides spiked with AQUA peptides, were introduced into the mass spectrometer via a stainless-steel Nano-bore emitter (OD 150 μm, ID 30 μm) with the spray voltage of 2 kV and capillary temperature 275 °C. The Orbitrap Fusion was operated in parallel reaction monitoring (PRM) mode. MS2 precursors were isolated with a quadrupole mass filter set to a width of 1.2 $m/z$. Precursors were fragmented by higher energy collision dissociation (HCD) and detected in Orbitrap detector with a resolution of 60,000. The normalized collision energy (NCE) for HCD was 25%. The values for the AGC target and maximum injection time were $5 \times 10^4$ and 118 ms, respectively. The peptide inclusion list is shown in Supplementary Table 4. Five μL (0.2 μg/μL) of tryptic peptides (spiked with AQUA peptides) of each trypsin-digested CSF sample, was injected and run in triplicates. Pooled CSF samples were run as quality control samples throughout all the samples.

The linear range of each heavy peptide was measured in pooled CSF digest. A mixture of heavy AQUA peptides was spiked at five different concentrations into pooled CSF samples. The concentration range was adjusted for each individual peptide according to its expected endogenous signal. Each sample was analyzed three times, and the peak areas of heavy peptides were plotted against the spiked heavy peptides concentrations (Fig. 5).

All the raw data generated on the Fusion MS were imported to Skyline v4.1 (MacCoss Lab Software, USA) for data analysis. Peak integration was done automatically by the software and was manually inspected to confirm correct peak detection. Peak identities were confirmed by measured transitions (dotp) and also between endogenous and corresponding heavy peptides (rdotp). The best 3–5 transitions were selected for the quantification and was performed by matching light and heavy peak area ratios. All calculations were performed using Microsoft Excel. For quality control, we also measured a peptide shared for C4A and C4B (tC4; GRIP). Finally, two subjects did

not display detectable C4A (in the GRIP chort) and C4B (in the KaSP cohort) peptide levels. We decided to initially exclude these subjects (one sample from the KaSP cohort and one sample from the GRIP cohort) in the main analyses. However, elevated levels of CSF C4A in FEP-SCZ remained in the GRIP cohort when including all values (Kruskal-Wallis $H$ test followed by post-hoc tests: adjusted $P$[FEP-SCZ vs. HCs] = 0.038, and adjusted $P$[FEP-SCZ vs. FEP-nSCZ] = 0.00003), while C4B levels in the KaSP cohort remained similar across groups (Kruskal–Wallis $H$ test followed by post-hoc tests: adjusted $P$[FEP-SCZ vs. HCs] = 0.905, and adjusted $P$[FEP-SCZ vs. FEP-nSCZ] = 0.193). Similar results were also obtained by imputing values with half of the limit of detection in the GRIP cohort (Kruskal–Wallis $H$ test followed by post-hoc tests: adjusted $P$[FEP-SCZ vs. HCs] = 0.039, and adjusted $P$ [FEP-SCZ vs. FEP-nSCZ] = 0.00002) and in the KaSP cohort (Kruskal-Wallis $H$ test followed by post-hoc tests: adjusted $P$[FEP-SCZ vs. HCs] = 0.859, and adjusted $P$[FEP-SCZ vs. FEP-nSCZ] = 0.151).

### Molecular analyses of *C4* structural elements (ddPCR)
Using ddPCR, *CN*s of *C4* structural elements (*C4A*, *C4B*, *C4-HERV* CNs) in the GRIP cohort were measured from genomic DNA isolated from blood samples[6]. *C4A-HERV+*, *C4A-HERV−*, *C4B-HERV+*, and *C4BHERV−* *CN*s were determined based on imputations. A detailed protocol for imputations can be found on Github: https://github.com/freeseek/imputec4. For distributions of *C4 CN*s in the samples see Supplementary Fig. 10. *C4A*, or *C4B*, *CN*s displayed no association to age (*C4A CN*: $r$[Spearman correlation] = 0.008 CI: −0.30–0.31; $P$ = 0.959, and *C4B CN*: $r$[Spearman correlation] = −0.03 CI: −0.33–0.28; $P$ = 0.852) or sex (*C4A CN*: $r$[Spearman correlation] = 0.08 CI: −0.23–0.38; $P$ = 0.602, and *C4B CN*: $r$[Spearman correlation] = 0.16 CI: −0.16–0.44, $P$ = 0.315).

### Molecular analyses of *C4* structural elements (imputation from WGS)
*CN*s of *C4* structural elements (*C4A-HERV+*, *C4A-HERV−*, *C4B-HERV+*, and *C4B-HERV−* *CN*s) in the KaSP cohort were imputed from MHC genotypes computed from whole genome sequencing (WGS) data. HapMap3 CEU reference haplotype panel, from Sekar et al.[1], was used to impute pre-annotated *C4* haplogroups, using Beagle (version 3.3), with subsets of SNPs in the extended MHC locus (chr6: 25–34 Mb).

### Cognitive testing
The Measurement and Treatment Research to Improve Cognition in Schizophrenia Consensus Cognitive Battery was used to evaluate cognitive function in the KaSP cohort. This battery contains 10 tests that measure 7 cognitive domains: Speed of processing (Brief Assessment of Cognition in Schizophrenia: Symbol Coding, Category Fluency: Animal Naming, Trail Making Test: Part A); Attention/vigilance (Continuous Performance Test-Identical Pairs); working memory (Wechsler Memory Scale-3rd Edition: Spatial Span, Letter-Number Span); verbal learning (Hopkins Verbal Learning Test-Revised); Visual learning (Brief Visuospatial memory Test-Revised); reasoning and problem solving (Neuropsychological Assessment Battery: Mazes) and social cognition (Mayer–Salovey–Caruso Emotional Intelligence Test: Managing Emotions). One psychologist (HFB) administered all the tests.

### Sensorimotor gating
Electromyography (EMG) was recorded from the orbicularis oculi using two 20-mm disk Ag/AgCl electrodes positioned below and lateral to the eye. A 40 × 50 mm ground electrode was implanted behind the ear on the bone. EMG activity was filtered (1- to 1000-Hz notch filter and 60-Hz notch filter) and digitized at a rate of 1 kHz. Acoustic stimuli were supplied via headphones using a Psylab Stand Alone monitor and a tone generator (containing a digital white noise source and a digitally controlled sine wave generator with a range of 30–2000 Hz from Contact Precision Instruments). The EMG signal was amplified with a

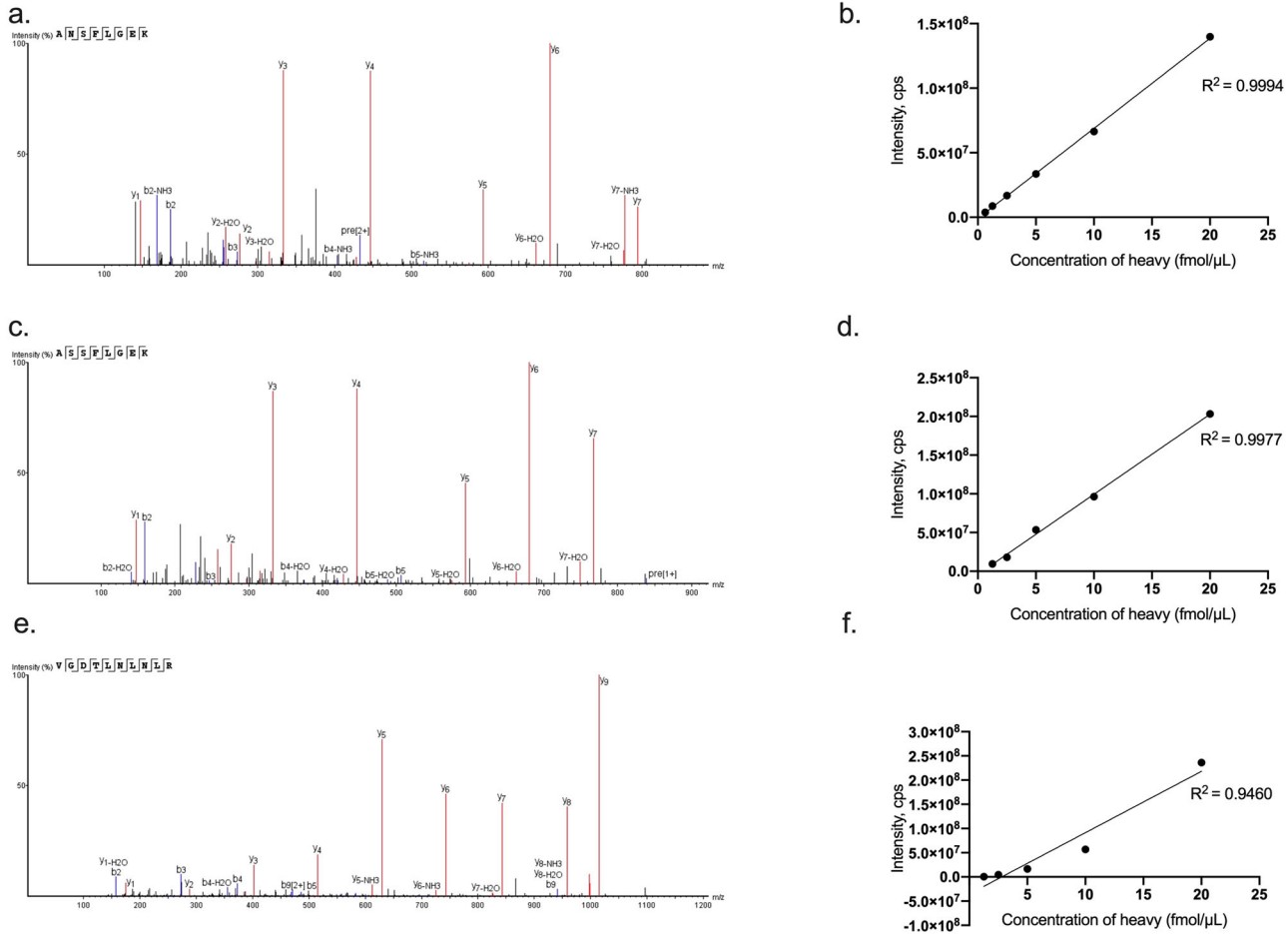

**Fig. 5 | Representative validation spectra and linearity graphs for C4 peptides.** Validation spectra for the unique peptides used to detect C4A (**a**), C4B (**c**) and total C4 (tC4) (**e**) in human cerebrospinal fluid (CSF). **b**, **d**, **f** display linear range of heavy peptides in pooled CSF digest for C4A, C4B, and tC4, respectively. Spiked pooled CSF samples were analyzed three times at each concentration. Each data point on the graph represents the mean of the three values. Source data are provided in the Source Data file.

Grass A.C. Amplifier (model 1CP511, Astro-Med., Inc.) and acquired by using commercially available hardware/software (BioPac) on a laptop computer (Hewlet Pacard Compaq 6715b).

All subjects (KaSP) were tested using the same startle system. Subjects were instructed to relax and keep their eyes open and directed against a position marked on the wall, during the test. Throughout the session, background noise was set to 70 dB. The session consisted of 60 trials with two conditions: pulse-alone trials, consisting of a 40 ms, 115-dB pulse of white noise and prepulse-pulse trials, consisting of a 40 ms 115 dB pulse (white noise, as above) preceded by a 16-dB (above background) prepulse of 20 ms duration (white noise). The ISI between the prepulse and the pulse tested were 30, 60, and 120 ms, respectively. A 3-min acclimation period with 70 dB white noise started the session. The immediately following block consist of 15 pulse-alone trials, used for the calculation of habituation using the latent curve modeling[37]. The next three blocks each consisted of 15 trials: 4–5 pulse-alone trials and 5 prepulse-pulse trials with the ISI of interest and 5 prepulse-pulse trials with other ISIs (not included in the analysis of PPI for the ISI of interest). Processing of the EMG recording was done offline with a 100-Hz high-pass filter and baseline correction by using a 100 ms prestimulus baseline. Response onset was defined by the first crossing from baseline within a 20–120 ms window after stimulus onset. The difference of the most positive peak and most negative in a 20–150 ms window after pulse onset was used to calculate the peak response amplitude. The following startle measures were examined; (1) reactivity, or the

magnitude of response (startle amplitude (SA), (2) PPI or the percentage of change in startle magnitude to prepulse + pulse versus pulse-alone trials ((pulse − prepulse + pulse) / pulse) * 100), (3)[29] habituation or the decrement in reactivity with repeated stimulus administration. The primary outcome measures of PPI were calculated for 30, 60, and 120 ms ISI conditions, respectively, as percent change in response amplitudes [% PPI = (mean startle alone- mean prepulse-pulse condition)/mean startle alone × 100]. Startle response was calculated by comparing the mean startle magnitude of the middle five pulses during the first trial (that only contained pulse alone).

### iPSC generation and neuronal differentiation
iPSCs from four chronic patients (all white males, 28 years old, 44 years old, 55 years old, and 58 years old, respectively) were used in this study. The patients were recruited from outpatient units at the Massachusetts General Hospital (MGH). All individuals signed a written informed consent before participating in the study, as approved by the Institutional Review Board of Partners HealthCare. All relevant ethical regulations were followed when performing the study. A structured psychiatric evaluation was performed using the Structured Clinical Interview for DSM-IV and the Mini-International Neuropsychiatric Interview by a single psychiatrist with at least 5 years of clinical experience[6].

Following subcutaneous injection of lidocaine 1%, a 3.0 or 4.0 mm punch tool was used to obtain a dermal biopsy sample from the non-dominant forearm. Biopsies were collected into a

transport medium of DMEM (Gibco) with 1% penicillin–streptomycin–glutamine (Corning Life Sciences, Tewksbury, MA, USA). Upon receipt of samples, they were washed twice in PBS with 10% penicillin–streptomycin–glutamine. Two 60-mm tissue culture-treated plates were used, one to cut up the biopsy and the other one to plate the pieces of biopsy. A scalpel was used to cut the biopsy into small pieces and then to grid the dish where the pieces of biopsy were plated (dermis side down). The dish was placed in an incubator at 37° for 15 min in order for the skin to adhere to the dish. Medium of DMEM with 10% FBS and 1% penicillin–streptomycin–glutamine was finally gently added over the pieces of skin.

The collected fibroblasts were then reprogrammed (mRNA reprogramming) and the resulting iPSC colonies stabilized and expanded under xeno-free conditions by Stemiotics (San Diego, CA, USA). Briefly, stable iPSCs were expanded in NutriStem XF medium (Biological Industries) and on biolaminin 521 LN-coated (BioLamina) plates to at least passage 3. iPSCs were then purified using MACS with anti-TRA-1-60 MicroBeads (Miltenyi Biotec) on LS columns according to the manufacturer's instructions. All fibroblasts and iPSCs were screened and found negative for Mycoplasma; they stained positive for octamer-binding transcription factor 4 (POU domain, class 5, transcription factor 1) and TRA-1-60 (Fig. 3a).

We then generated *NGN2* expressing stable NPC lines using TALEN-based plasmids as previously described[6]. Briefly, the doxycycline-inducible *NGN2* AAVS1 knock-in plasmid, based on an AAVS1 SA-2A-puro-pA donor (plasmid no. 22075; Addgene), was generated by replacing the puromycin resistance gene with the neomycin resistance gene and cloning a cassette containing the Tet-On 3G promoter driving the human *NGN2* complementary DNA followed by P2A, a zeocin resistance gene, the BGH polyA, CMV early enhancer/chicken β actin (CAG) promoter driving Tet-On 3G and BGH polyA. The AAVS1-targeting TALENs are based on hAAVS1 1R TALEN and hAAVS1 1L TALEN (plasmid nos. 35432 and 35431, respectively; Addgene) and were generated by golden gate assembly. Plasmid maps can be provided upon request. NPCs were collected with Accutase and counted. For each transfection, $4 \times 10^6$ cells were pelleted at 300 g and then resuspended in 100 μL prewarmed nucleofector solution (Human Stem Cell Kit 1, Lonza, catalog no. VPH-5012); 4 μg of *NGN2* plasmid and 1.5 μg each of 1R and 1L plasmids were added directly to the resuspended cells, or negative control without plasmid, followed by nucleofection (Amaxa Nucleofector II; Lonza) according to the manufacturer's protocol using program B-016. After nucleofection, cells were plated onto Geltrex-coated six-well plates in neural expansion medium (50% NBM, 50% advanced DMEM/F-12 with 1× NIS). Stable lines were expanded by selection with the addition of 125 μl mL$^{-1}$ G418 (Geneticin; Thermo Fisher Scientific) over 8–10 d or until the negative transfection control cells completely died.

*NGN2* neurons were then generated as described earlier[6]. Briefly, NPCs were plated as single cells on poly-L-ornithine/laminin coated 24-well polypropylene plates at $2.5 \times 10^5$ cells/500 μL in 50% neurobasal medium (NBM; ThermoFisher Scientific) and 50% DMEM/F12 (ThermoFisher Scientific) with 1× neural induction supplement (Thermo Fisher Scientific). On the following day, the medium was replaced with neuronal differentiation medium (50% NBM and 50% DMEM/F12 supplemented with 1× MEM Non-essential Amino Acids (Thermo-Fisher Scientific), 1× N-2 supplement (ThermoFisher Scientific), 10 ng mL$^{-1}$ brain-derived neurotrophic factor (BDNF; Peprotech, Cat# 450-02) and 10 ng m L$^{-1}$ recombinant human NT-3 (Peprotech, Cat# 450-03)). Day 2 cells were differentiated in neuronal differentiation medium enriched with 1× B-27 (ThermoFisher Scientific) and 2 μg m L$^{-1}$ doxycycline (Sigma-Aldrich, Cat# D3073). This medium was maintained for the entire differentiation process, unless noted. On days 3–6, cells were incubated with the same medium with the addition of 5 μg m L$^{-1}$ zeocin (ThermoFisher Scientific). For neuronal maturation, 11 days after seeding, astrocyte conditioned medium (ACM: ScienCell Research Laboratories, Cat# SC1811-sf) was added to the culture. Assays were then performed at day 25.

## Immunocytochemistry

*NGN2* neurons were cultured on cover slides coated with poly-L-ornithine/laminin. Cells were fixed in 4% paraformaldehyde in PBS for 15 min. Following permeabilization in PBS containing 0.1% Triton-X100 for 5 min, the slides were blocked with blocking solution (PBS containing 5% normal goat serum) for 1 hour at room temperature. Primary antibodies against MAP-2 (1:500, Abcam, Cat# ab5392), Beta III Tubulin (1:2000, Promega, Cat# G7121), Oct4 (1:400, Cell signaling, Cat# 2840), TRA 1-60 (1:500, Abcam, Cat#Ab16288) in blocking solution were added and incubated overnight at 4 degrees C. Cells were then washed with PBS and incubated with the corresponding Alexa Fluor-conjugated secondary antibodies (1:500) for 1 h at room temperature. The following secondary antibodies were used: goat anti-chicken (ThermoFisher Scientific, Cat# A32933), goat anti-mouse (Abcam, Cat# ab150113) and goat anti-rabbit (Abcam, Cat# A32732. Images were obtained using a confocal microscope system (Zeiss, LSM800).

## Cytokine stimulations of *NGN2* neurons and *C4A*/*C4B* mRNA expression

After 25 days, mature *NGN2* neurons were treated with either VEH, 1 ng/mL IL-1β (Thermo Fisher Scientific, cat# PHC0815) or 1 ng/mL IL-6 (Thermo Fisher Scientific, cat# PHC0063), alone or in combination for 24 hours. Following stimulations of cells, total RNA was extracted using Trizol® Reagent (Invitrogen) according to the manufacturer's protocol. The quantity and purity of total RNA was measured using a NanoDrop® 1000 spectrophotometer (Thermo Scientific). Complementary DNA was synthesized using High-Capacity RNA-to-DNA™ Kit (Applied Biosystems) following the manufacturer's protocol. qPCR was performed with TaqMan Gene Expression Assays (Applied Biosystems) by using the StepOnePlus Real-time PCR system (Applied Biosystems). Each 20 μL PCR reaction contained 5 μL of cDNA, 300 nM of forward and reverse primer, 250 nM of probe and 10 μL of TaqMan PCR master mix (Applied Biosystems). Primers for *C4A* and *C4B* reactions were as follows (forward: GCAGGAGACATCTAACTGGCTTCT and reverse: CCGCACCTGCATGCTCCT). Specific probes for C4A (FAM-ACC CCT GTC CAG TGT TAG-MGB) and C4B (FAM-ACC TCT CTC CAG TGA TAC-MGB) were used. ΔCt values were obtained by comparison to Ct values of the housekeeping gene (GAPDH, Assay ID. Hs99999905_m1) for each technical replicate. Fold changes (2 − ΔΔCt) were then generated by normalization to the average ΔCt value for VEH treated wells per plate. *C4B* mRNA could not be detected in non-stimulated cultures from one subject (not genotyped) and these cultures were excluded from quantification of *C4B* mRNA expression after stimulations. In one culture, *C4A* and *C4B* mRNA expression were also ~100× larger than in the other cultures and this well was excluded (see Source data). All other wells that passed QC (ICC, morphology) were analyzed by qPCR and included in the statistical analyses.

## CSF cytokine analyses

IL-1beta and IL-6 were previously measured as part of the Novex Human Ultrasensitive Cytokine 10-Plex Panel (product number at that time: LHC6004; now distributed by Invitrogen) on the Luminex immunoassay platform. In contrast to the manual provided with the kit, an additional calibration curve point was included to extend the assay dynamic range in the low concentration range. 50 μl sample was diluted with 50 μl of sample diluent, the final sample dilution factor was 1:2. The assay was processed as recommended by the kit instructions. The measurement was performed on a FM3D instrument (operated under software version 4.0, Luminex, Austin, TX, USA). Data analysis was performed using MasterPlex QT 2010 (V 2.0, Hitachi Software

Engineering, San Francisco, CA, USA). Standard curves were generated using weighted $(1/y^2)$ 5-parameter logistics. The analyte concentrations in the samples were calculated based on the standard curves. In this study, only IL-1beta and IL-6 were initially analyzed (based on our in vitro data). Upon request from one of the reviewers, data on all cytokines were then extracted from the database for post-hoc analyses. Seven more cytokines were detected in more than 10 subjects (IL-1beta, IL-6, IL-10, IL-5, TNFα, GM-CSF, and IL-8) but none of them displayed a significant association with C4A (Supplementary Table 5).

### CSF neuronal pentraxins analyses

Preparation of samples and LC-MS/MS analyses were performed as previously described[20] using micro-high-performance liquid chromatography-mass-spectrometry (6495 Triple Quadrupole LC/MS system, Agilent Technologies). Briefly, in a flowrate of 0.3 mL/min, sample injection of 40 μL and a 30 min gradient was employed for separation on a Hypersil Gold reversed phase column (dim. $100 \times 2.1$ mm, particle size 1.9 μm, Thermo Fisher Scientific). The two mobile phases (A and B) were composed of 0.1% formic acid in water (v/v) and 0.1% formic acid/84% acetonitrile in water (v/v), respectively. Electrospray settings were the following: gas temperature 220 °C, gas flow 15 L/min, nebulizer 40 psi, sheath gas temperature 200 °C, sheath gas flow L/min, capillary voltage in positive mode 3500 V and, nozzle voltage at 500 V. iFunnel settings were the following in positive mode: high-pressure RF 200 V and low-pressure RF 160 V. Individually optimized collision energies and cell accelerator voltages were used for each peptide. The scheduled multiple reaction monitoring method used a delta retention time of 0.8 min for each peptide. Quality control samples, consisting of pooled CSF samples obtained from the Neurochemistry Laboratory at Sahlgrenska University Hospital, Mölndal, Sweden, were injected at regular intervals to monitor the performance of the assay over time. The collection and use are in accordance with the Swedish law on biobanks in healthcare (2002:297). Skyline 20.1 (MacCoss Lab Software) was used for peak visualization and inspection. Peak adjustment was performed if required for optimal peak area calculation. Relative peptide concentrations were obtained by summing all measured fragment peak areas for each peptide and dividing that by the sum of the fragment peak areas of the corresponding IS, followed by multiplication of the amount of IS added per μL of CSF.

### Statistics

Available sample size for the initial analysis in the discovery cohort was deemed adequate based on the data presented by Sekar et al.[6] (an estimated correlation coefficients of at least 0.5 for *C4A* (or *C4B*) CNs versus corresponding RNA expression in 101 subjects, and an odds ratio of 1.4 ($P = 2 \times 10^{-5}$) comparing *C4A* RNA expression in 35 SCZ subjects and 70 HCs). According to the pre-defined analytical plan, a significant increase in C4A (or C4B) concentration in the discovery cohort ($P < 0.05$, two-sided p-value) was to be followed-up in an independent cohort then again measuring C4A, as well as C4B, with significance set at $P < 0.05$ (two-sided *p*-value). Potential confounders were to be tested on group levels (in each cohort). If this indicated any evidence of between group differences (on a liberal p-value threshold of 0.1), indicated variables were to be studied in relation to CSF C4A or C4B concentration. If this revealed an association with $P < 0.05$, we were to perform an adjusted analysis. The exception to this rule was (1) antipsychotics, here we before analyses decided to compare levels between medicated and non-medicated patients, and even if we observed no indication of altered levels in medicated vs. unmedicated we were to perform stratified/adjusted analyses, and (2) as post-hoc analyses, we also performed adjusted analyses (sex, age, and smoking) to control for possible residual confounding. For group comparisons, CSF C4A and C4B concentrations (or *C4A* and *C4B* RNA in experiments) in the main analyses were analyzed using Kruskal-Wallis *H* tests followed by post-hoc tests. Continuous variables were analyzed by

Spearman's or Pearson's correlation analyses (or partial correlation analyses) as indicated. Experimental data were analyzed using Kruskal–Wallis *H* test followed by post-hoc tests. Data were analyzed using R for Mac OS (version 4.2.1), IBM SPSS statistics (version 28.0.0.0), and Graphpad prism 9 for macOS (version 9.3.1).

### Reporting summary

Further information on research design is available in the Nature Research Reporting Summary linked to this article.

## Data availability

Source data, necessary to interpret and verify the research in the article, are provided in the Source Data file. In accordance with the institutional regulations and the Swedish law, raw data containing sensitive information that can be used to identify individuals, cannot be shared publicly under the current data protection. Instead, such data can be made available upon request (to the corresponding author) and on a case-by-case basis as allowed by the legislation and ethical permits. Source data are provided with this paper.

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

## Acknowledgements

We thank the patients who contributed to this study. Support from the Swedish National Infrastructure for Biological Mass Spectrometry (BioMS) is gratefully acknowledged. We are thankful to Steven A. McCarroll lab (Harvard medical School) for support with imputing *C4* structural elements from WGS data. C.M.S is supported by grants from the Swedish Research Council, One Mind Foundation, Marianne and Marcus Wallenberg Foundation, and Erling-Persson Family Foundation. F.O received support from the Swedish Brain Foundation and the Swedish Society for Medical Research. KB is supported by the Swedish Research Council (#2017-00915), the Alzheimer Drug Discovery Foundation (ADDF), USA (#RDAPB-201809-2016615), the Swedish Alzheimer Foundation (#AF-742881), Hjärnfonden, Sweden (#FO2017-0243), the Swedish state under the agreement between the Swedish government and the County Councils, the ALF agreement (#ALFGBG-715986), the European Union Joint Program for Neurodegenerative Disorders (JPND2019-466-236), the National Institute of Health (NIH), USA, (grant #1R01AG068398-01), and the Alzheimer's Association 2021 Zenith Award (ZEN-21848495). H.Z. is a Wallenberg Scholar supported by grants from the Swedish Research Council, the European Research Council, and the Swedish Federal Government under the LUA/ALF agreement. NMI received financial support from the State Ministry of Baden-Wuerttemberg for Economic Affairs, Labour and Tourism. S.E. received financial support from the Swedish Research Council. The St. Göran study is supported by grants from the Swedish Research Council (M.L.), the Swedish Foundation for Strategic Research (M.L.), the Swedish Brain Foundation (M.L.), and the Swedish Federal Government under the LUA/ALF agreement (M.L.). KaSP is supported by grants from the Swedish Federal Government under the LUA/ALF agreement (C.M.S., S.E., and S.C.). All illustrations in Figs. 1, 2 and 3 were created with Biorender.

## Author contributions

J. Gracias and C.M.S. contributed to the overall design, direction, analysis, and reporting of the study. F.O., J. Gracias, and C.S.M. designed in vitro experiments. J. Gracias, F.O., S.G., T.F., İ.Ş.D., and N.K. derived neural cultures and performed the in vitro cytokine assay and qPCR. S.D.S. and R.H.P. provided the NGN2 stable neural progenitor cells for the in vitro experiments. C.P.G. performed the ddPCR for the GRIP cohort. S.M. extracted WGS data from KaSP cohort for *C4A* and *C4B*. J.Gr. imputed predicted *C4A* RNA levels from this data. M.S. and C.M.S. supervised the genetic analyses. J.H.L. performed shotgun mass spectrometry and helped to develop the method to detect C4A and C4B in CSF. L.S. helped collect and interpret the PPI and habituation data. L.S., S.E., G.E., S.C., F.P., H.F.B., and C.M.S. are part of the KaSP project and helped recruit and phenotype subjects and collect CSF. H.F.B. performed the cognitive testing of patients in the KaSP cohort. K.B., E.H., H.Z., M.L., A.P., A.G., K.A., and A.I are part of the GRIP consortium and were involved in recruiting and phenotyping subjects and collecting CSF. J.N. and A.B. analyzed KaSP CSF for neuronal pentraxins. V.C. and J.C.G. analyzed CSF cytokine levels. All authors discussed the results and implications and commented on the manuscript at various stages.

## Funding

## Competing interests

The authors declare no competing interests.

## Additional information

[1]Department of Physiology and Pharmacology, Karolinska Institutet, Stockholm, Sweden. [2]The Institute of Neuroscience and Physiology, University of Gothenburg, Gothenburg, Sweden. [3]Psychosis Clinic, Sahlgrenska University Hospital, Mölndal, Sweden. [4]Center for Quantitative Health, Center for Genomic Medicine and Department of Psychiatry, Massachusetts General Hospital, Boston, MA, USA. [5]Department of Clinical Neuroscience, Karolinska Institutet, Stockholm, Sweden. [6]Stockholm Health Care Services, Region Stockholm, Stockholm, Sweden. [7]Department of Anesthesiology, Sahlgrenska University Hospital, Gothenburg, Sweden. [8]Novartis Institutes of BioMedical Research, Cambridge, MA, USA. [9]Department of Psychology, University of Gothenburg, Gothenburg, Sweden. [10]Department of Molecular Medicine and Surgery, Karolinska Institutet and Center for Molecular Medicine, Karolinska University Hospital, Stockholm, Sweden. [11]NMI Natural and Medical Sciences Institute at the University of Tübingen, Reutlingen, Germany. [12]Clinical Neurochemistry Laboratory, Sahlgrenska University Hospital, Mölndal, Sweden. [13]Department of Neurodegenerative Disease, UCL Institute of Neurology, Queen Square, London, UK. [14]UK Dementia Research Institute at UCL, London, UK. [15]Hong Kong Center for Neurodegenerative Diseases, Hong Kong, China. [16]Department of Medical Sciences, Psychiatry, Uppsala University, Uppsala, Sweden. [17]Department of Medical Epidemiology and Biostatistics, Karolinska Institutet, Stockholm, Sweden. ✉e-mail: carl.sellgren@ki.se

