## [Peer Review File · Nature Communications]

Cerebrospinal fluid concentration of complement component 4A is increased in first episode schizophreniaReviewers' comments:

Reviewer #1 (Remarks to the Author):

The authors have conducted a very interesting study following up on the previous results from Sekar et al., where data on in vivo protein levels have been lacking. The particular advantage of this study is the human CSF analysis, where they show differences in C4A for individuals with schizophrenia.

Specific points

- How sure are we that "Excessive synapse loss is a core feature of schizophrenia" as stated in the abstract?
- The abstract doesn't contain any estimates and confidence intervals on the primary outcome
- Please give estimate and confidence intervals in the text also of the relative and absolute risk and the risk difference between the groups, so the reader can better get a sense of if it is a small or large difference – particularly for the studies on human CSF, which are of greatest importance
- Maybe subheadings highlighting when the results are from human and when they are from animal studies could improve the readability of the manuscript
- Spelling mistake page 11 line 222 "fassed"
- Line 223 is "non-cutting" the right word?
- It is a bit unclear if the CSF samples were tested individually or only as pooled samples
- It is unclear if the human CSF were selected from larger cohorts or if this was the total CSF samples taken from the cohorts
- Line 373 "as previously described previously" – delete the last previously
- Is the iPSC only from 1 patient? If so it should be highlighted as a major limitation regarding the generalization
- In the method section CSF analysis of 10 cytokines are described, but the results and correlation with C4 is not clearly described for all of them, and for how many they could measure which cytokines in the CSF
- Figure 1: P-values are not sufficient and the focus should be on the estimates and confidence intervals
- It is not clear what the primary primary outcome was before doing the analysis - if it was overall differences and if the authors thought C4A would also be increased in psychotic disorders to some extent. The rationale for when adjusting for multiple testing could be clearly written.
- It seems to be smaller differences between the groups which should be highlighted
- In the text it says that the estimates are means but should it be medians instead?
- So when there seem to be a stronger correlation between C4A and IL1, is it then IL1 – what are the directions of causality and how do it fit in the context of the Sekar study? And again only p-values are listed and it seems as the direction is similar for IL6 but just not significant – and what about the remaining 8 cytokines?
- Are IL1 also correlated with C4A in healthy controls?
- Figure 1D – it seems as the levels are lower for FEP-aSCZ than HC and then higher for FEP-SCZ than controls – how does this fit in? Do the FEP-SCZ have more inflammatory activity than the other groups?
- The healthy controls are they super healthy or how is the comorbidity patterns of general medical conditions in the two groups?
- supp tabl 1: Antipsychotic use in HC should be listed to 0 if correct, which would also give a difference between groups on this
- should the analysis be adjusted for age?
- the analysis adjusted for antipsychotic medication, could also be performed stratified for antipsychotic use to get an impression if the FEP-nSCZ group on antipsychotic medication are different from FEP-SCZ on antipsychotic medication

Reviewer #2 (Remarks to the Author):

This paper relates on very important physiopathological aspect of first episode psychosis (FEP) which is the role of complement system in psychosis, based on the hypothesis that excessive synaptic pruning induced by excessive complement activity would be one of the core mechanism underlying psychosis.

The study is original, based on CSF analysis of FEP which are rare studies due to the difficulty of getting such samples, and on a new way to measure the Complement protein.

The method is totally adequate, although it would have been interesting to look for correlation with brain imaging data to explore if there is any link between levels of complement and reduced cortical thickness.

The work clearly supports the hypothesis and the conclusions made.

Reviewer #3 (Remarks to the Author):

This is a very good study, with strong convergent evidence. The use of multiple independent cohorts to ensure reproducibility is to be lauded.

One suggestion for improvement is to discuss how IL1B may be involved in synaptic pathology in (co-morbid) disorders other than schizophrenia. Specifically, the work by Niculescu and colleagues (Le-Niculescu et al. 2013, Niculescu et al 2015) has identified IL1B as a biomarker for suicidality. One could speculate that the cognitive impairment that underlies suicidality (either in terms of impulsivity or impaired decision making) may have a similar IL1B-C4 pathway.

In fact, in the future one could conduct such follow-up studies or secondary analyses in CSF samples from this or related cohorts.

Reviewer #4 (Remarks to the Author):

This is an interesting pilot study using a modest size of clinical samples.

Clinical samples are relatively small, and the definition of FEP-nSCZ is unclear. Are they patients with schizophreniform or schizoaffective or bipolar with psychotic features or what? How about the ethnic group? All Caucasians? Premorbid IQ? Education duration? Several details are missing.

Many groups have studied C4 in CSF in schizophrenia, and one group was already published (Gallego et al, 2021). This reviewer cannot find this paper cited in this manuscript, but it is very important to discuss with that paper. Multiple other groups are also in the process for publications on C4 and schizophrenia (although this reviewer has never searched these in preprint repositories, there may be some). C4 in CSF has been studied also in multiple other neuropsychiatric conditions, and what is the specificity of C4 elevation in schizophrenia?

By using only one patient with this syndrome (schizophrenia), interesting preliminary results are shown. The study of CSF shows some supportive data to these interesting results from cell culture. How many clones of iPS cells from one subject are tested? No detail is available. Given that patients with schizophrenia are very heterogenous with each other, any experiment with only one patient is regarded as very preliminary. By using these preliminary but interesting data as an entry point, a scientific mechanistic study will be arranged and such a study will contribute to the field in the future.

If the authors wish to publish this manuscript as a short report of a clinical study, the details that are required for clinical papers may be amplified in a more rigorous manner. If the authors wish to publish this manuscript as a manuscript of a multi-disciplinary study, more mechanistic insight beyond clinical correlative observations will be expected. In summary, this reviewer expects the present, preliminary but interesting study in a specialized journal in psychiatry.

We are thankful for the constructive comments provided by the reviewers. Most importantly, we have now performed the in vitro experiments using iPSC lines from a total of four patients. A complete point-by-point response to reviewers' suggestions is detailed below.

Reviewers' comments:

Reviewer #1 (Remarks to the Author):

The authors have conducted a very interesting study following up on the previous results from Sekar et al., where data on in vivo protein levels have been lacking. The particular advantage of this study is the human CSF analysis, where they show differences in C4A for individuals with schizophrenia.

Specific points

- How sure are we that "Excessive synapse loss is a core feature of schizophrenia" as stated in the abstract?

We have now rephrased to "Postsynaptic density is reduced in schizophrenia, and risk variants increasing complement component 4A (C4A) gene expression leads to excessive synapse elimination in patient-derived schizophrenia models". Line 37-39.

- The abstract doesn't contain any estimates and confidence intervals on the primary outcome **This is now added.**

- Please give estimate and confidence intervals in the text also of the relative and absolute risk and the risk difference between the groups, so the reader can better get a sense of if it is a small or large difference – particularly for the studies on human CSF, which are of greatest importance

We agree with the reviewer and have added correlation coefficients with CIs for all correlation analyses. For Kruskal-Wallis H test we report medians and CIs for each group. In the main text we report post-hoc p-values for the important comparisons, and in figure legends we report all p-values.

- Maybe subheadings highlighting when the results are from human and when they are from animal studies could improve the readability of the manuscript

We fully agree. The manuscript was initially sent to Nature Neuroscience as a brief report and then transferred to Nature Communication. Thus, the manuscript was formatted according to those recommendations, as well as complying with size restrictions. As suggested, the manuscript has been re-formatted to include subheadings, expanding the discussion etc.

- Spelling mistake page 11 line 222 "fassed" **Thanks for noticing this. It is now corrected.**

- Line 223 is "non-cutting" the right word?

Yes, it is correct (also called a blunt or atraumatic needle; this is now added). Atraumatic needles decrease the risk of post-dural-puncture headache substantially compared to cutting/sharp needles.

- It is a bit unclear if the CSF samples were tested individually or only as pooled samples

The CSF samples were tested individually. This is now added to MS (Page 17; line 373 and line 380).

- It is unclear if the human CSF were selected from larger cohorts or if this was the total CSF samples taken from the cohorts

“All eligible subjects with available data were included in the present study” (page 15, line 334, page 16, line 366). The main limiting factor was availability of CSF. Two subjects with CSF did however not display detectable CSF levels of C4A (page 20, line 450). One of these subjects had ddPCR data and no C4A CNs (line 461). We decided to initially exclude these subjects from the main analyses (line 451-452) but we then also performed sensitivity analyses including them with zero values, as well as imputing values with half of the limit of detection, and with similar results (line 452-454).

- Line 373 “as previously described previously” – delete the last previously
Thanks for noticing this. It has been corrected.

- Is the iPSC only from 1 patient? If so it should be highlighted as a major limitation regarding the generalization

It is correct that repeated experiments on one iPSC line (patient) was used. In these experiments, we were focusing on cytokine induced expression of C4A as a general mechanism rather than comparing differences between patients and controls. Thus, we used one patient (and repeated experiments) with a representative genotype (2 CNs each of C4A and C4B) to make the comparison between C4A and C4B easier. However, we agree that this is a limitation and we have now generated data from a total of four SCZ patients (see Page 9; line 191-192 and Figure 2b and 2c) then showing similar results. As the relative increase in mRNA expression by cytokine induction may vary with different CNs, we still believe the data on the line with equal numbers of C4A and C4B CNs is important, thus we also repeated these experiments on this line together with the experiments on the new lines (page 9, line 198-203).

- In the method section CSF analysis of 10 cytokines are described, but the results and correlation with C4 is not clearly described for all of them, and for how many they could measure which cytokines in the CSF

It is correct that we extracted data for IL-1 β and IL-6 from a 10-Plex Panel. Only IL-1 β and IL-6 were analyzed (as these cytokines were the ones we selected to analyze in the *in vitro* model). The Göpfert lab has now kindly provided all data, and 7 cytokines could be detected in more than 10 subjects (IL-1 β , IL-6, IL-10, IL-5, TNF α , GM-CSF, and IL-8). Of these cytokines, three independent clusters could be identified based on $r < 0.5$ and $P > 0.05$. Except for IL-1 β , none of the cytokines displayed a significant (or close to significant) association with C4A (either unadjusted or adjusted by a Bonferroni correction). See page 26; lines 609-612 and page 27; lines 613-616.

- Figure 1: P-values are not sufficient and the focus should be on the estimates and confidence intervals

As suggested below, we now analyze this data using Kruskal-Wallis H test (non-parametric). We plot CIs, give absolute values, and in figure legends we report all test statistics.

- It is not clear what the primary primary outcome was before doing the analysis - if it was

overall differences and if the authors thought C4A would also be increased in psychotic disorders to some extent.

This is an excellent question. We have now clarified that our main hypothesis was that FEP patients converting to SCZ would have higher C4A levels than HCs, as well as FEP patients not converting to SCZ (given that *e.g.*, PRS has been used to discriminate FEP-SCZ from FEP-nSCZ). The FEP-nSCZ group then serves as a highly matched negative control group to the FEP-SCZ group. This way we could, to some degree, control for confounders that inevitably will exist when a patient cohort is compared to a sex- and age-matched group of healthy controls, such as exposure to other medications, diet, activity levels etc. See page 5; lines 79-87. Notably, we still choose to use two-sided p-values in all analyses despite the clear hypothesis.

The rationale for when adjusting for multiple testing could be clearly written.

We agree and have now added the pre-defined analytical plan to the statistical section (page 28). In sum, potential confounders were tested on group levels (in each cohort). If this indicated any evidence of between group differences (on a liberal p-value threshold of 0.1), indicated variables were to be studied in relation to CSF C4A or C4B concentration. If this revealed a significant association, we performed an adjusted analysis.

- It seem to be smaller differences between the groups which should be highlighted
In the combined cohort, we observed a 57% increase (as compared to HCs). We have added that the increase is still relatively modest (page 11; line 237-241).

- In the text it says that the estimates are means but should it be medians instead?
It is correct that the estimates previously reported were means of log-transformed values (and normalized to control groups in graphs) as this data fulfilled the criteria of the used statistical models. However, we have now changed ANOVAs to Kruskal Wallis *H* tests (as well as using spearman's correlation coefficient instead of Pearson's) and display medians with CIs. As expected, the results were similar.

- So when there seem to be a stronger correlation between C4A and IL1, is it then IL1 – what are the directions of causality and how do it fit in the context of the Sekar study? And again only p-values are listed and it seems as the direction is similar for IL6 but just not significant
This is an important comment. If IL-1 β /IL-6 induces C4A mRNA expression (and IL-1 β and IL-6 are increased in SCZ) then C4A CNs should not fully explain the observed C4A RNA up-regulation in SCZ brain material. This is also exactly what has been observed (*e.g.*, Kim *et al.*, Nat. Neurosci. 2021 Jun;24(6):799-809). This is now more clearly described in the revised manuscript (page 8; lines 171-177 and page 9; lines 178-180).

In CSF, with both IL-1 β and IL-6 present (*i.e.*, most closely mimicking the experimental condition using the IL-1 β /IL-6 stimulation), we only observe a significant association between IL-1 β and C4A. This data suggests, that in an *in vivo* context, variance in IL-1 β levels have a greater influence on C4A protein levels per C4A CN (*i.e.*, lowering IL-1 β levels is likely a more effective strategy to decrease C4A levels). However, as pointed out by the reviewer, it is important to emphasize that our mechanistic data clearly indicate that both are needed to induce C4A mRNA expression per C4A CN (and that most available strategies to decrease IL-1 β levels would also influence IL-6 levels). In the

revised manuscript, we now more clearly discuss that the mechanism depends on IL-1 β and IL-6 while CSF this mechanism is best captured by variance in IL-1 β levels.

– and what about the remaining 8 cytokines?

Please see response above.

- Are IL1 also correlated with C4A in healthy controls?

This was not studied (no such data was available) and a potential patient vs. control difference would much likely require a very large sample size. However, our hypothesis was that the mechanism is not specific to SCZ. Instead, it would contribute to increasing the patient vs. control difference in C4A levels per C4A CN as SCZ patients in general display higher concentrations of these cytokines in CSF. This is now added to the discussion (page 11; line 244-247)

- Figure 1D – it seems as the levels are lower for FEP-aSCZ than HC and then higher for FEP-SCZ than controls – how does this fit in? Do the FEP-SCZ have more inflammatory activity than the other groups?

It is correct that in the GRIP cohort, FEP-nSCZ patients displayed a median C4A concentration below HCs. As also indicated by the p-value, we believe this was due to randomness. Combining the both cohorts, medians for HCs and FEP-nSCZ were also very similar (HCs; n=41; Median=0.29 fmol/ul: CI=0.29-0.39 and FEP-nSCZ; n=28; Median=0.28 fmol/ul: CI=0.25-0.32), even more so when considering C4A levels per C4A CN). Thus, our data support a selective increase in C4A levels in FEP-SCZ patients. This is partly due C4A CNs although a significant elevation in C4A protein levels remain after adjusting for genetically predicted C4A RNA expression. Our mechanistic data show that IL-1 β /IL-6 can increase neuronal C4A RNA expression per C4A CN, and the correlation between IL-1 β and C4A in CSF suggest that this mechanism is relevant (and likely targetable) in vivo. This indirectly suggests, that as a group, FEP-nSCZ patients could have slightly lower IL-1 β /IL-6 levels as compared to FEP-SCZ patients. To the best of our knowledge, no such studies have been performed (due to low power and lack of follow-up data, patient vs. control analyses are prioritized). While it indirectly can be predicted from our data that median IL-1 β levels are higher in FEP-SCZ as compared to FEP-nSCZ, our data set was not powered to detect a significant increase in IL-1 β in FEP-SCZ vs. FEP-nSCZ, even if IL-1 β /IL-6 would exclusively explain the increase in C4A unrelated to genetically predicted C4A RNA expression. Thus, such an analysis done here would be inconclusive and must await larger samples.

- The healthy controls are they super healthy or how is the comorbidity patterns of general medical conditions in the two groups?

In the discovery cohort, controls were recruited by advertisement, and we used same exclusion criteria as for patients. It is to be expected that they are healthier than the patients, however, the difference is likely much smaller than if we would have compared with chronic SCZ patients. In contrast, the controls in the replication cohort were randomly selected by Statistical Sweden with the same exclusion criteria as in the patient cohort and therefore less likely to differ regarding comorbidity and life-style factors. Notably, despite our efforts to find adequately matched controls, we acknowledge the potential of residual confounding (page 5; lines 93-95) and therefore judged it prudent to include an FEP-nSCZ group. Please see “Study populations and diagnostic assessments” for more details.

- supp tabl 1: Antipsychotic use in HC should be listed to 0 if correct, which would also give a difference between groups on this

We have now added statistics using Fisher's exact test.

- should the analysis be adjusted for age?

No significant differences were observed between groups regarding age. Thus, according to our pre-defined analysis plan we did not correct for age (see Statistics). However, we have now added a post-hoc analysis studying the effect of age on C4A and C4B levels. As expected, adjustment for age did not change our results. Controlling for age, C4A CSF in FEP-SCZ is still significantly increased compared to FEP-nSCZ ($p < 0.001$) and HC ($p = 0.008$). C4B CSF is still not significantly different in FEP-SCZ compared to FEP-nSCZ ($p > 0.999$) and HC ($p = 0.405$).

- the analysis adjusted for antipsychotic medication, could also be performed stratified for antipsychotic use to get an impression if the FEP-nSCZ group on antipsychotic medication are different from FEP-SCZ on antipsychotic medication

This is a good suggestion. CSF C4A levels are significantly higher in FEP-SCZ than FEP-nSCZ also when only including patients on an antipsychotic agent (as well as when only including antipsychotic naïve patients). See Page 7, line 129-136 and Supplementary Figure 1 and 2.

Reviewer #2 (Remarks to the Author):

This paper relates on very important physiopathological aspect of first episode psychosis (FEP) which is the role of complement system in psychosis, based on the hypothesis that excessive synaptic pruning induced by excessive complement activity would be one of the core mechanism underlying psychosis.

The study is original, based on CSF analysis of FEP which are rare studies due to the difficulty of getting such samples, and on a new way to measure the Complement protein. The method is totally adequate, although it would have been interesting to look for correlation with brain imaging data to explore if there is any link between levels of complement and reduced cortical thickness.

The work clearly supports the hypothesis and the conclusions made.

We are grateful to the reviewer for these encouraging words. We agree that looking at correlations between CSF C4A levels and MRI data would be highly interesting. In the current study we focused on CSF synapse markers (in previous studies strongly correlated with PFC thickness; Van Der Ende *et al.* 2020, ref. nr.19) as very few participants had MRI and CSF C4A data (and requirement of multiple testing adjustments for MRI measurements in numerous brain regions would have rendered such an analysis severely under-powered).

Reviewer #3 (Remarks to the Author):

This is a very good study, with strong convergent evidence. The use of multiple independent cohorts to ensure reproducibility is to be lauded.

We are grateful to the reviewer for this encouraging words.

One suggestion for improvement is to discuss how IL1B may be involved in synaptic pathology in (co-morbid) disorders other than schizophrenia. Specifically, the work by Niculescu and colleagues (Le-Niculescu et al. 2013, Niculescu et al 2015) has identified IL1B as a biomarker for suicidality. One could speculate that the cognitive impairment that underlies suicidality (either in terms of impulsivity or impaired decision making) may have a similar IL1B-C4 pathway.

In fact, in the future one could conduct such follow-up studies or secondary analyses in CSF samples from this or related cohorts.

We thank the reviewer for these very good suggestions. Indeed, these papers are of relevance and is now discussed (page 11; line 254, and page 12; lines 252-255). The current patients are also included in a longitudinal cohort study, and we will take the reviewer's advice to look at suicidality over time when more follow-up data have been generated.

Reviewer #4 (Remarks to the Author):

This is an interesting pilot study using a modest size of clinical samples.

We thank the reviewer for these encouraging words. Notably, the study was designed with a discovery cohort (sample sized based on a power calculation), followed by a replication in an independent cohort of appropriate size. We respectfully disagree with the statement that sample sizes are modest, as this study uses prospective collection of CSF samples from acutely and carefully phenotyped psychotic first-episode patients who are unmedicated (or have medicated for a few weeks with antipsychotics). To the best of our knowledge, this is in fact the first study so far that includes a replication of a CSF finding in first-episode patients.

Clinical samples are relatively small, and the definition of FEP-nSCZ is unclear. Are they patients with schizophreniform or schizoaffective or bipolar with psychotic features or what? How about the ethnic group? All Caucasians? Premorbid IQ? Education duration? Several details are missing.

The definition of FEP-nSCZ as well as the (typical) diagnoses these subjects received at follow-up can be found in the manuscript. See page 14 (lines 320-323) and page 16 (lines 365-367). The demographic and clinical data (Supplementary Table 1 and 2) was limited in the previous version due to the format (and the fact that these cohorts have been thoroughly in many previous papers). We apologize for this and have now provide a comprehensive descriptive dataset including multiple rating scales, demographics, and socioeconomic factors (including data on education). While all participants in KaSP cohort were Caucasians, as reported, information on ethnicity was not included in the ethical approval for GRIP. Notably, analyses were adjusted for *C4A/C4B* CNs. Finally, FEP patients are included in the study the first time they display psychotic symptoms

and get in contact with health care. A cognitive evaluation using MATRICS is performed at this stage (Table 1) but IQ assessments before the study participants entered the study (i.e., premorbid IQ) is not available. This would require another study design.

Many groups have studied C4 in CSF in schizophrenia, and one group was already published (Gallego et al, 2021). This reviewer cannot find this paper cited in this manuscript, but it is very important to discuss with that paper. Multiple other groups are also in the process for publications on C4 and schizophrenia (although this reviewer has never searched these in preprint repositories, there may be some). C4 in CSF has been studied also in multiple other neuropsychiatric conditions, and what is the specificity of C4 elevation in schizophrenia? **Several studies of total C4 levels (likely to be indicative of a more general immune activation) have been performed in foremost neurodegenerative and autoimmune disorders. Our current data (as well as previous genetic and experimental work) clearly suggest that “the specificity of C4 elevation in schizophrenia” is the selective increase in C4A CNs, C4A RNA expression, and (now) C4A protein levels. To our best knowledge, this has not been observed in any other psychiatric or autoimmune disorders. This has now been more clearly stated in the revised version.**

We are thankful to the reviewer for bringing the recent paper by Gallego *et al.* to our attention. We now cite and discuss this paper in the revised manuscript (page 13; lines 276-285). In sum, this paper measures total C4 levels in CSF from 32 chronic schizophrenia spectrum disorder patients (SSD), all on an antipsychotic agent and with a median duration of illness of 14 years, together with 32 controls. The study has no replication cohort. Gallego *et al.* report that they see no differences in total CSF C4 levels between SSD patients in the unadjusted analysis. However, after performing an adjusted analysis (age and sex) they see an increase in total C4 levels in SSD. They stress the lack of genetic data and the inability to measure C4A/C4B levels as limitations. We have now also measured levels of total C4 levels (a peptide strongly correlating with C4A + C4B peptide levels) and could not replicate the finding by Gallego *et al.* despite a sample size almost twice as large. If the increase observed by Gallego *et al.* is not due to randomness (especially since this finding was not replicated), one of the most obvious differences between the studies is that they sample chronic patients on long-term antipsychotic medication instead of FEP patients. Possibly, C4A levels (or C4B levels) could increase in chronic patients, either due to pathophysiological events or related to confounders.

By using only one patient with this syndrome (schizophrenia), interesting preliminary results are shown. The study of CSF shows some supportive data to these interesting results from cell culture. How many clones of iPS cells from one subject are tested? No detail is available. Given that patients with schizophrenia are very heterogenous with each other, any experiment with only one patient is regarded as very preliminary. By using these preliminary but interesting data as an entry point, a scientific mechanistic study will be arranged and such a study will contribute to the field in the future.

This is an important issue raised by the reviewer. In these experiments we focused on cytokine induced expression of C4A as a general mechanism rather than comparing differences between patients and controls (i.e., we are not aiming to capture a SCZ unique mechanisms). This is now clarified (page 11; lines 243-249). Instead, we expect the mechanism to be more pronounced in SCZ as previous data indicate higher central cytokine levels in SCZ. We believe the experiments on a line with equal numbers of C4A

and C4B is important as IL-1 β /IL-6 induction could theoretically vary by CNs for the corresponding C4 gene (we have now performed three independent experiments using this line). As indirectly suggested by the reviewer, we have also now used lines from a total of 4 different subjects and the results are similar (Figure 2a-c).

If the authors wish to publish this manuscript as a short report of a clinical study, the details that are required for clinical papers may be amplified in a more rigorous manner. If the authors wish to publish this manuscript as a manuscript of a multi-disciplinary study, more mechanistic insight beyond clinical correlative observations will be expected. In summary, this reviewer expects the present, preliminary but interesting study in a specialized journal in psychiatry.

Given the possibility to expand the text (*i.e.*, original report instead of short report) we now include details as required for an observational clinical study. The experimental part is now also reported as to be expected. We believe that the combination of a rigorous clinical observation study (including replication in an independent cohort) with supporting experimental data is a rather common design in Nature Communications. For example, see the PET study (n=41) from Onwordi EC *et al.* Nat Commun. 2020;11.

Reviewers' comments:

Reviewer #1 (Remarks to the Author):

The authors have responded excellent to my prior comments.

I only have minor considerations remaining:

- Regarding potential confounders it would not in newer literature be commendable to only include in the adjustment models the ones indicated by evidence of group differences (as important potential confounders might not be significant particularly in smaller sample sizes), but instead adjusting for the potential confounders based on the literature – particularly age and sex should be adjusted for in the analyses as also indicated in sup Fig 6-7. And maybe smoking should be included in the adjustment models also.
- Could the authors show IL-1B median levels among HC, nSCZ and SCZ alongside the median level of C4a, for instance in bar charts, which might make it easier to see directly if the differences are similar, and the story then might be more on IL-1B, and not on synaptic pruning

Reviewers' comments:

Reviewer #1 (Remarks to the Author):

The authors have responded excellent to my prior comments.

I only have minor considerations remaining:

- Regarding potential confounders it would not in newer literature be commendable to only include in the adjustment models the ones indicated by evidence of group differences (as important potential confounders might not be significant particularly in smaller sample sizes), but instead adjusting for the potential confounders based on the literature – particularly age and sex should be adjusted for in the analyses as also indicated in sup Fig 6-7. And maybe smoking should be included in the adjustment models also.

This is a good comment. We have now included an adjusted analysis with sex, age, and smoking as covariates (see adjusted analysis in supplementary figure 1). In accordance with the similar age and sex distribution in the compared groups in both cohorts, we still observe a similar and significantly increase in C4A levels in FEP-SCZ patients while C4B levels were not elevated.

- Could the authors show IL-1B median levels among HC, nSCZ and SCZ alongside the median level of C4a, for instance in bar charts, which might make it easier to see directly if the differences are similar, and the story then might be more on IL-1B, and not on synaptic pruning

This is a good suggestion. Below, we have plotted CSF IL-1beta concentrations for FEP-SCZ and FEP-nSCZ patients (as mentioned before, CSF IL-1beta for the HCs were not available) together with CSF C4A concentrations (now also included as Supplementary Figure 5). CSF IL-1beta levels are higher in FEP-SCZ patients than in FEP-nSCZ patients but the relative increase is not as large as for CSF C4A levels. This supports the hypothesis that IL-1beta contributes to the elevated C4A levels observed in FEP-SCZ.